# One-Step Flow Q-Learning: Addressing the Diffusion Policy Bottleneck in Offline Reinforcement Learning

**Thanh Nguyen & Chang D. Yoo**
Department of Electrical Engineering
Korea Advanced Institute of Science and Technology (KAIST)
Daejeon, Korea
{thanhnguyen, cd_yoo}@kaist.ac.kr

## Abstract

Diffusion Q-Learning (DQL) has established diffusion policies as a high-performing paradigm for offline reinforcement learning, but its reliance on multi-step denoising for action generation renders both training and inference slow and fragile. Existing efforts to accelerate DQL toward one-step denoising typically rely on auxiliary modules or policy distillation, sacrificing either simplicity or performance. It remains unclear whether a one-step policy can be trained directly without such trade-offs. To this end, we introduce One-Step Flow Q-Learning (OFQL), a novel framework that enables effective one-step action generation during both training and inference, without auxiliary modules or distillation. OFQL reformulates the DQL policy within the Flow Matching (FM) paradigm but departs from conventional FM by learning an average velocity field that directly supports accurate one-step action generation. This design removes the need for multi-step denoising and backpropagation-through-time updates, resulting in substantially faster and more robust learning. Extensive experiments on the D4RL benchmark show that OFQL, despite generating actions in a single step, not only significantly reduces computation during both training and inference but also outperforms multi-step DQL by a large margin. Furthermore, OFQL surpasses all other baselines, achieving state-of-the-art performance in D4RL.

## 1 Introduction

In recent years, offline reinforcement learning (Offline RL) has achieved impressive progress through the integration of diffusion models, leading to many high-performance algorithms. A prominent example is Diffusion Q-Learning (DQL) (Wang et al., 2022), which replaces the conventional diagonal Gaussian policy in TD3-BC (Fujimoto & Gu, 2021) with a denoising diffusion probabilistic model (DDPM) (Ho et al., 2020). This approach has demonstrated substantial performance gains and has spurred widespread interest in leveraging generative models for policy learning. Notably, DQL remains competitive and often outperforms many more recent methods in both diffusion-based planning and policy optimization (Lu et al., 2025a; Dong et al., 2024).

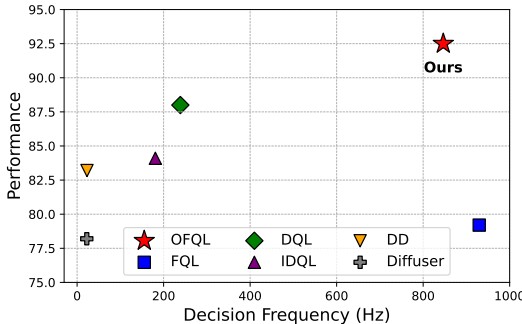

Figure 1: **Performance and decision frequency.** Performance (*i.e.,* normalized score) and decision frequency are measured on an A100 GPU and averaged across MuJoCo tasks from D4RL. OFQL achieves both high inference speed and strong performance, clearly outperforming prior baselines.

Despite its strong empirical results, DQL faces key practical limitations, including high computational demands during training and inference (Kang et al., 2023; Wang et al., 2022), as well as optimization fragility causing reduced performance (Park et al., 2025). Upon closer analysis, we identify that the

key bottleneck lies in its use of DDPM diffusion policy (Ho et al., 2020), which involves multiple denoising steps per action, leading to slow inference. Furthermore, the training speed is doubly affected: beyond the diffusion loss, DQL requires two rounds of policy sampling per iteration—one for the current action and another for the next—to compute all loss components. In addition, DQL leverages the reparameterization trick to backpropagate through the entire denoising chain, amplifying computational load and impeding convergence to optimal solutions. These characteristics collectively hinder DQL's efficiency and robustness.

It is worth noting that several recent approaches have partially addressed these limitations—through improved denoising solvers (Kang et al., 2023), IQL-based learning (Hansen-Estruch et al., 2023), or the use of an auxiliary policy and policy distillation strategies (Park et al., 2025; Chen et al., 2023; 2024; Lu et al., 2025b). Nevertheless, such solutions typically introduce additional complexity, multi-phase training procedures, or undesirable trade-offs in scalability and policy quality.

Recognizing that the diffusion policy itself is the bottleneck, we adopt a more direct approach by introducing One-Step Flow Q-Learning (OFQL), a novel framework specifically designed to enable effective one-step action generation during both training and inference, without the need for auxiliary models, policy distillation, or multi-stage training. At the heart of OFQL is the elimination of DDPM's computationally intensive multi-step denoising and associated reparameterization trick. By recasting DQL policy under the Flow Matching (Lipman et al., 2022) paradigm, we facilitate its efficient action sampling. However, conventional Flow Matching frequently yields curved trajectories, limiting one-step inference accuracy—an issue rooted in the intrinsic properties of the marginal velocity field it models. We address this by learning an average velocity field instead, enabling accurate direct action prediction from a single step. As a result, OFQL eliminates the necessity of iterative denoising and recursive gradient propagation, providing a faster, more stable, one-step training-inference pipeline. Extensive empirical evaluations on the D4RL benchmark demonstrate that OFQL not only surpasses DQL in performance but also significantly improves both training and inference efficiency, all while maintaining a simple learning pipeline. Compared to other approaches, OFQL delivers consistently stronger results, establishing it as a fast and state-of-the-art algorithm on D4RL.

## 2 RELATED WORK

**Diffusion Models in Offline Reinforcement Learning.** Offline RL aims to learn effective policies from fixed datasets without environment interaction (Levine et al., 2020), with early methods addressing distributional shift via conservative objectives—e.g., CQL (Kumar et al., 2020), TD3+BC (Fujimoto & Gu, 2021), and IQL (Kostrikov et al., 2021). However, these approaches often rely on unimodal Gaussian policies, which struggle to model complex, multi-modal action distributions. To address this, recent approaches have adopted diffusion models (Ho et al., 2020; Song et al., 2020b) for offline RL. These models excel in representing complex distributions and have been applied in various forms: as planners for trajectory generation (Janner et al., 2022; Ajay et al., 2022), as expressive policy networks (Wang et al., 2022; Hansen-Estruch et al., 2023), and as data synthesizers to augment training (Zhu et al., 2023).

Among diffusion-based methods, Diffusion Q-Learning (DQL) (Wang et al., 2022) stands out as a strong baseline, replacing Gaussian policies in TD3+BC with a diffusion model to better capture multi-modal actions. Follow-up evaluations, including those by Clean Diffuser (Dong et al., 2024) and recent empirical studies (Lu et al., 2025a;b), confirm DQL's consistent advantage over policy-based and planner-based methods. Despite its effectiveness, DQL remains significant computational overhead during both training and inference (Kang et al., 2023), and is prone to suboptimal convergence (Park et al., 2025).

Subsequent works have attempted to mitigate these issues. For instance, early approaches use an efficient solver that reduces the number of denoising steps (Kang et al., 2023). Other approaches bypass backpropagation through time (BPTT) of the DQL policy update by training a diffusion policy to clone the behavior policy, with actions reweighted by a separately learned IQL-based value function (Hansen-Estruch et al., 2023). Others further apply distillation to obtain a one-step policy (Chen et al., 2023; 2024). However, IQL-based methods are generally less effective than actor–critic learning (Park et al., 2025). To address this, (Park et al., 2025) adopts a flow model to clone the behavior policy and distill it into a one-step policy for actor–critic updates. While yielding a one-step inference

policy, the distillation process still requires repeated queries to the underlying multi-step diffusion or flow model. Overall, although these methods partially alleviate DQL's limitations, they introduce additional components and multi-phase training procedures, thereby increasing system complexity and limiting practical generality. Departs from prior work, our method pinpoints the diffusion policy itself as the primary source of inefficiency and instability in DQL. We directly solve it with a highly performant one-step policy alternative that provides a simpler and more robust solution—without relying on auxiliary policies, distillation, or sacrificing policy expressivity.

**Efficient One-Step Diffusion.** Our work is inspired by advances in efficient generative modeling with diffusion models and flow-based models (Sohl-Dickstein et al., 2015; Ho et al., 2020; Song & Ermon, 2019). To accelerate generation, one line of work focuses on distillation techniques that compress multi-step models into fewer steps (Salimans & Ho, 2022; Sauer et al., 2024; Yin et al., 2024), while another pursues flow matching approaches that learn time-dependent velocity fields for straight-through sampling (Lipman et al., 2022; Liu et al., 2022). Consistency Models (Song et al., 2023) offer another path to one-step generation, but suffer from training instabilities (Song & Dhariwal, 2023; Lu & Song, 2024). More recent methods (Frans et al., 2024; Geng et al., 2025; Zhou et al., 2025) address these limitations by exploiting different physical parametrization using dual time variables, showing improved stability and performance. Leveraging the Mean Flow modeling (Geng et al., 2025), OFQL realizes a one-step flow-based policy while uniquely incorporating the Q-gradient to guide velocity learning, instead of relying solely on supervised learning.

## 3 PRELIMINARIES

### 3.1 OFFLINE RL.

Reinforcement learning (RL) is typically formalized as a Markov Decision Process (MDP), defined by $\mathcal{M} = (\mathcal{S}, \mathcal{A}, P, R, \gamma)$. Here, $\mathcal{S}$ and $\mathcal{A}$ denote the state and action spaces, $P(s' \mid s, a)$ denotes the transition probability of moving from $s$ to $s'$ given action $a$, and $R(s, a)$ denotes the reward function. The discount factor $\gamma \in [0, 1)$ governs the trade-off between immediate and future rewards. In RL, the objective is to learn a policy $\pi_\theta(a \mid s)$, parameterized by $\theta$, that maximizes the expected discounted return $\mathbb{E}_\pi \left[ \sum_{h=0}^{\infty} \gamma^h R(s_h, a_h) \right]$.

To support policy learning, the action-value (Q) function is defined:

$$Q^\pi(s, a) = \mathbb{E}_\pi \left[ \sum_{h=0}^{\infty} \gamma^h R(s_h, a_h) \mid s_0 = s, a_0 = a \right], \tag{1}$$

which measures the expected cumulative return starting from state $s$ and action $a$ under policy $\pi$.

Offline RL is a setting of RL where the agent does not interact with the environment but instead learns from a fixed dataset of transitions $\mathcal{D} = \{(s_h, a_h, s_{h+1}, r_h)\}$. The challenge lies in learning an optimal policy solely from this static dataset that often contains suboptimal behavior, without any further exploration.

### 3.2 DIFFUSION Q-LEARNING (DQL)

**Modeling Policy as a Diffusion Model.** The Denoising Diffusion Probabilistic Model (DDPM) (Ho et al., 2020) is a powerful generative framework that formulates a forward diffusion process as a fixed Markov chain, progressively corrupting data into noise, and learns a parametric reverse process to reconstruct the data. Once trained, DDPM is capable of generating complex data distributions by reversing the diffusion process, starting from pure noise.

Viewing actions as data, (Wang et al., 2022) formulate the reverse process of a DDPM, conditioned on state $s$, as a parametric policy $\pi_\theta$:

$$\pi_\theta := p_\theta \left( a^{0:K} \mid s \right) = p(a^K) \prod_{k=1}^{K} p_\theta \left( a^{k-1} \mid a^k, s \right), \tag{2}$$

where $a^K \sim \mathcal{N}(\mathbf{0}, \mathbf{I})$ and $a^0$ denotes the clean action corresponding to the actual policy output. For training, following DDPM (Ho et al., 2020), DQL parameterizes the Gaussian distribution

$p_\theta(a^{k-1}|a^k, s)$ with the variance fixed as $\boldsymbol{\Sigma}(a^k, k; s) = \beta^k \boldsymbol{I}$ where the $\{\beta^k\}_{k=1}^K$ are predefined variance schedule values and the mean is defined via a noise prediction model:

$$\boldsymbol{\mu}_\theta(a^k, k; s) = \frac{1}{\sqrt{\alpha^k}}\left(a^k - \frac{\beta^k}{\sqrt{1 - \bar{\alpha}^k}}\epsilon_\theta(a^k, k; s)\right), \tag{3}$$

where $\alpha^k = 1 - \beta^k$, $\bar{\alpha}^k = \prod_{i=1}^k \alpha^i$, and $\epsilon_\theta$ is a neural network predicting Gaussian noise.

To enforce behavior cloning, the score matching loss can be used as a training objective. Specifically, the model minimizes:

$$\mathcal{L}_{\text{DBC}}(\theta) = \mathbb{E}_{k,\epsilon,(a^0,s)\sim\mathcal{D}}\left[\left\|\epsilon - \epsilon_\theta\left(\sqrt{\bar{\alpha}^k}a^0 + \sqrt{1 - \bar{\alpha}^k}\epsilon, k; s\right)\right\|^2\right], \tag{4}$$

where $k \sim \mathcal{U}\{1, \ldots, K\}$, $(a^0, s)$ are sampled from the offline dataset $\mathcal{D}$, and $\epsilon \sim \mathcal{N}(\boldsymbol{0}, \boldsymbol{I})$ denotes Gaussian noise.

Once trained, to generate an action (i.e., $a^0$), the model sequentially samples from $K$ conditional Gaussians, starting from $a^K \sim \mathcal{N}(\boldsymbol{0}, \boldsymbol{I})$:

$$a^{k-1} = \frac{1}{\sqrt{\alpha^k}}\left(a^k - \frac{\beta^k}{\sqrt{1 - \bar{\alpha}^k}}\epsilon_\theta(a^k, k; s)\right) + \sqrt{\beta^k}\epsilon, \tag{5}$$

**Behavior-regularized actor-critic.** To form a complete offline RL algorithm, DQL adopts the behavior-regularized actor–critic framework (Wu et al., 2019; Fujimoto & Gu, 2021), alternating between minimizing the actor and critic losses.

Specifically, the critic loss, which focuses on training the Q network, is defined as:

$$\mathcal{L}(\phi) = \mathbb{E}_{(s,a,r,s')\sim\mathcal{D}, a'\sim\pi_{\theta'}}\left[\left(r + \gamma \min_i Q_{\phi_i'}(s', a') - Q_{\phi_i}(s, a)\right)^2\right], \tag{6}$$

where $i \in 1, 2$ indexes the two Q networks for double Q-learning, and $(\phi', \theta')$ denote target network parameters updated via exponential moving average (EMA) (Fujimoto & Gu, 2021). The actor loss, which focuses on learning the policy, is defined as:

$$\mathcal{L}(\theta) = \mathcal{L}_{\text{DBC}}(\theta) - \alpha \cdot \mathbb{E}_{s\sim\mathcal{D},a\sim\pi_\theta}\left[Q_\phi(s, a)\right], \tag{7}$$

where $\alpha$ is the weighting coefficient. To normalize for dataset-specific Q-value scales, $\alpha$ is adapted as $\alpha = \frac{\eta}{\mathbb{E}_{(s,a)\sim\mathcal{D}}[\|Q_\phi(s,a)\|]}$, with $\eta$ being a tunable hyperparameter. The denominator is treated as a constant for optimization.

Despite being implemented with a relatively simple MLP architecture for both the diffusion policy and Q-functions, DQL has demonstrated strong performance across standard offline RL benchmarks such as D4RL (Fu et al., 2020), outperforming many recent diffusion-based policies and planners (Dong et al., 2024; Lu et al., 2025a;b).

Nevertheless, DQL exhibits two notable limitations: (1) slow training and inference (Kang et al., 2023), and (2) susceptibility to unstable or suboptimal training (Park et al., 2025).

## 4 RATIONALE AND METHODOLOGY

The slow inference time stems from its reliance on a denoising diffusion process, where actions are sampled through a reverse chain of $K$ Gaussian transitions (Eq. 5). Due to the Markovian nature of the DDPM framework, sampling an action during inference requires the same number of denoising steps $K$ as those used during training. Additionally, a large $K$ is typically necessary to ensure that $a^K$ approximates an isotropic Gaussian. Reducing $K$ breaks this assumption, often resulting in significant performance degradation.

In training, DQL also exhibits compounding inefficiencies. First, the critic loss (Eq. 6) requires sampling target actions $a' \sim \pi_{\theta'}$, each of which must be generated via the full $K$-step denoising, introducing considerable computational overhead. Enforcing one-step action generation, however, can destabilize training due to diffusion sampling errors. Second, the actor loss (Eq. 7) requires sampling actions from $\pi_\theta$ and performing backpropagation through all $K$ denoising steps (i.e., BPTT)

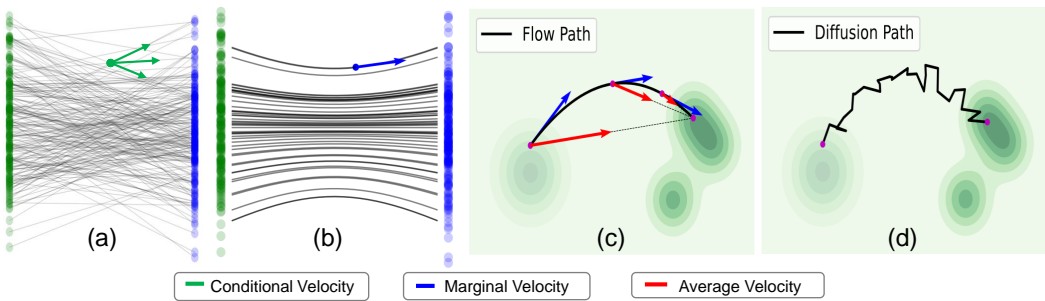

Figure 2: Comparison between diffusion and flow matching. (a) Conditional flows arise from different $(\epsilon, x)$ pairs, resulting in varying conditional velocities. (b) Marginal velocity is obtained by averaging over these conditional velocities. (c) Flow paths are inherently curved, but average velocity fields enable direct one-step transport from noise to data. (d) Diffusion paths are also curved but noisy, making one-step denoising challenging. Note that all the velocities exhibit symmetry under time reversal. As the model is trained to parameterize the forward flow (from data to noise), inference inverts this direction to generate samples. Accordingly, for clarity, we plot the negative velocity vector to represent the reverse generation trajectory.

using the reparameterization trick (Eq. 5). Although this enables end-to-end training, the recursive gradient flow through a long stochastic computation graph is well-known to be prone to numerical instability and can potentially lead to suboptimal outcomes (Chen et al., 2023; Park et al., 2025).

At first glance, diffusion policy sampling appears to be the bottleneck, but resolving DQL's limitations is far from straightforward. For example, using an efficient solver, e.g., the DDIM solver (Song et al., 2020a), could reduce denoising steps, yet in our experiments, applying DDIM for one-step action generation severely degraded policy performance. Similarly, replacing diffusion with consistency models, such as Consistency-AC (Ding & Jin, 2023), still requires multiple denoising steps and yields lower performance. The most effective one-step approaches are distillation-based methods (Park et al., 2025; Chen et al., 2024; 2023; Lu et al., 2025b), which accelerate inference through student policies but incur an additional training phase or shift the inefficiencies to the distillation stage. This raises a natural question: Can we design a one-step policy that directly eliminates inefficiencies in both training and inference?

**Designing One-Step Policy.** Diffusion models generate samples through stochastic and often curved trajectories, which makes one-step sampling challenging. Flow Matching (FM) (Lipman et al., 2022) offers a principled alternative by mapping noise directly to data along smoother, more direct paths (Liu et al., 2022), as illustrated in Figure 4 (c,d). Modeling the policy via flow matching can potentially improve both efficiency and stability. Let us model the policy as a variant of flow matching based on linear paths and uniform time sampling. Given data $a, s \sim \mathcal{D}$ and noise $\epsilon \sim \mathcal{N}(\mathbf{0}, \boldsymbol{I})$, FM defines a linear flow path $a_t$ and *conditional velocity* $v_t(a_t | a; s)$ for a particular $s$ as :

$$a_t = (1 - t)a + t\epsilon, \quad v_t = \frac{da_t}{dt} = \epsilon - a, \quad \text{where} \quad t \in [0, 1] \tag{8}$$

Note that by formulation, $a \equiv a_0$, and we use $a$ without a subscript to denote the clean action for simplicity.

FM essentially learns the *marginal velocity*, parametrized by the neural network $v_\theta(a_t, t; s)$, using the Conditional Flow Matching loss:

$$\mathcal{L}_{\text{CFM}}(\theta) = \mathbb{E}_{t \sim U[0,1], (a,s) \sim \mathcal{D}, \epsilon} \left\| v_\theta(a_t, t; s) - v_t(a_t \mid a; s) \right\|^2. \tag{9}$$

Once trained, sampling an action for a state $s$ proceeds by solving the ODE $\frac{da_t}{dt} = v(a_t, t; s)$, starting from $a_1 \equiv \epsilon \sim \mathcal{N}(\mathbf{0}, \boldsymbol{I})$, approximated using a solver such as Euler's method: $a_{t-\Delta t} = a_t - \Delta t \cdot v(a_t, t; s)$.

Intuitively, since flows are designed so that the overall transport trajectory becomes approximately straight (Liu et al., 2022), one might expect that modeling the policy via Flow Matching could support one-step generation by setting $\Delta t = 1$. However, in practice, the sampling trajectory is straight only when the target distribution collapses to a delta distribution or when rectification or similar techniques are explicitly applied. Without such conditions, the marginal velocity field typically induces a curved

overall trajectory, preventing reliable one-step action generation. Importantly, the curvature is not simply a consequence of imperfect neural approximation, but rather an inherent property of the ground-truth marginal velocity field. This phenomenon is illustrated in Figure 2 (a,b,c).

To enable high-quality one-step generation, we reinterpret the velocity field $v(a_t, t; s)$ in Flow Matching as the *instantaneous velocity*, and instead propose to model the *average velocity*, which directly connects any two arbitrary time steps. Specifically, we define the average velocity over an interval $[r, t]$ as:

$$u(a_t, r, t; s) \triangleq \frac{1}{t - r} \int_r^t v(a_\tau, \tau; s) \, d\tau, \tag{10}$$

representing the total displacement across the interval divided by its duration. Here, $r$ and $t$ denote the target and current times, respectively, with the constraint $0 \leq r \leq t \leq 1$.

In general, the average velocity is a functional of the instantaneous velocity, *i.e.*, $u = \mathcal{F}[v]$. This field $u$ is fully determined by the instantaneous velocity field $v$ and is independent of any neural network. We therefore treat $u$ as the ground-truth average velocity field and train a neural network $u_\theta$ to approximate it using a loss, referred to as the *Average-Velocity Matching* loss:

$$\mathcal{L}_{\text{FBC}^\star}(\theta) = \mathbb{E}_{0 \leq r \leq t \leq 1; \, s, \epsilon} \left[ \left\| u_\theta(a_t, r, t; s) - u(a_t, r, t; s) \right\|^2 \right]. \tag{11}$$

Once $u_\theta$ is learned, actions can be generated in a single step through the approximate endpoint map

$$a = T_\theta(\epsilon, s) = \epsilon - u_\theta(\epsilon, r{=}0, t{=}1; s), \qquad \epsilon \sim \mathcal{N}(0, I), \tag{12}$$

which eliminates the iterative ODE integration required by standard Flow Matching. This avoids both the computational overhead and the discretization error associated with numerical ODE solvers. A formal justification for why this one-step procedure preserves the FM action accuracy is provided in Appendix G.

As a result, the learned policy $\pi_\theta(a \mid s)$ is the push-forward of the Gaussian prior through the approximate endpoint map:

$$\pi_\theta = (T_\theta)_\# \mathcal{N}(0, I).$$

Moreover, when used as a regularizer, optimizing the average-velocity matching loss $\mathcal{L}_{\text{FBC}^\star}(\theta)$ encourages the learned one-step policy $\pi_\theta(\cdot \mid s)$ to remain close to the behavior policy $\mu(\cdot \mid s)$, effectively performing behavior cloning. Importantly, this behavior cloning still preserves the ability to model complex, multimodal action distributions through the nonlinear transport map inherited from Flow Matching (see Appendix H for a formal justification).

**Practical Loss.** In practice, computing $u$ from its definition requires integration, which is computationally intractable for optimization. To address this, we adopt an equivalent reformulation based on the *MeanFlow Identity* (Geng et al., 2025):

$$u(a_t, r, t; s) = v(a_t, t; s) - (t - r) \frac{d}{dt} u(a_t, r, t; s), \tag{13}$$

where the total derivative expands as $\frac{d}{dt} u(a_t, r, t; s) = v(a_t, t; s) \cdot \partial_{a_t} u + \partial_t u$. The computation of the derivative also remains efficient by leveraging Jacobian–vector products. Notably, when $t = r$, the target $u$ reduces to the instantaneous velocity.

Equation 13 is mathematically equivalent to Eq. 10 (the detailed derivation is provided in Appendix O). We therefore use the Eq. 13 to compute the average velocity $u_{\text{tgt}}$, avoiding explicit integration in Eq. 10. The resulting field can be fitted by a neural network using the following loss:

$$\mathcal{L}_{\text{FBC}}(\theta) = \mathbb{E}_{t, r, r \leq t, (a, s) \sim \mathcal{D}, \epsilon} \left\| u_\theta(a_t, r, t; s) - \text{sg}(u_{\text{tgt}}) \right\|_2^2. \tag{14}$$

In this loss, $a_t = (1 - t)a + t\epsilon$. The $v(a_t, t; s)$ in Eq. 13 is additionally replaced with the conditional velocity $v_t$, following FM, to approximate the instantaneous velocity on the fly during training. Consequently, the target velocity is defined as:

$$u_{\text{tgt}} = v_t - (t - r)\big(v_t \cdot \partial_{a_t} u_\theta + \partial_t u_\theta\big), \tag{15}$$

where $v_t = \epsilon - a$. The operator $\text{sg}(\cdot)$ denotes stop-gradient, preventing higher-order gradients on the target during optimization.

**One-step Flow Q-Learning.** To form OFQL, we model the policy with the average velocity parameterization $u_\theta$. Its behavior regularization loss is $\mathcal{L}_{\text{FBC}}(\theta)$, defined in Eq. 14. In addition, since the target average velocity is computed based on the estimate of the instantaneous velocity (Eq. 13), accurate estimation of the instantaneous velocity is crucial for effective average-velocity learning. To encourage this, when sampling $(t, r)$ we enforce a certain *flow ratio* $\lambda$, i.e., the probability that $t = r$. This design biases training toward learning the instantaneous velocity, while still allowing regression to the average velocity, improving the bootstrapping.

Given the policy $u_\theta$, sampling an action becomes a differentiable one-step operation:

$$a \sim \pi_\theta(\cdot \mid s) \quad \Leftrightarrow \quad a = \epsilon - u_\theta(\epsilon, r = 0, t = 1; s), \quad \text{where} \quad \epsilon \sim \mathcal{N}(0, I) \qquad (16)$$

The critic and actor losses are updated as:

$$\mathcal{L}(\phi) = \mathbb{E}_{(s,a,r,s')\sim\mathcal{D},\, a'\sim\pi_{\theta'}}\left[\left(r + \gamma \min_{i\in\{1,2\}} Q_{\phi_i'}(s', a') - Q_{\phi_i}(s, a)\right)^2\right],$$

$$\mathcal{L}(\theta) = \mathcal{L}_{\text{FBC}}(\theta) - \alpha\, \mathbb{E}_{s\sim\mathcal{D},\, a\sim\pi_\theta}\left[Q_\phi(s, a)\right]. \qquad (17)$$

With the new one-step policy, the main modification from DQL losses is that the behavior regularization term and action sampling require only a single step.

## 5 EXPERIMENT SETTING

**Benchmarks.** We evaluate OFQL on a diverse set of tasks from the D4RL benchmark suite (Fu et al., 2020), a widely adopted standard for offline reinforcement learning. Our evaluation spans various domains, including locomotion, navigation and manipulation tasks to demonstrate the method's generalizability. Detailed task descriptions and experimental protocols are provided in Appendix A.

**Baselines.** To rigorously assess OFQL's performance, we compare it against a broad spectrum of representative baselines, categorized as follows: (1) Non-Diffusion policies: Behavior Cloning (BC), TD3-BC (Fujimoto & Gu, 2021), and IQL (Kostrikov et al., 2021); (2) Diffusion-based planners: Diffuser (Janner et al., 2022), Decision Diffuser (DD) (Ajay et al., 2022); (3) Multi-step Diffusion-based policies: IDQL (Hansen-Estruch et al., 2023), DQL (Wang et al., 2022), and EDP (Kang et al., 2023); and (4) One-step Flow policies: FQL (Park et al., 2025)

**Implementation Details.** Our approach builds directly on DQL (Wang et al., 2022), inheriting its training and inference procedures to ensure a fair comparison. We adopt the original DQL architecture for both the Q-function and policy networks. The only minor modification lies in the policy input, which is augmented by concatenating an additional positional embedding corresponding to the target step $r$, in addition to the standard timestep embedding $t$. For timestep sampling, the $t$ and $r$ are sampled from a logit-normal distribution (Esser et al., 2024) with parameters $(-0.4, 1.0)$. The main hyperparameters are the *flow ratio* and $\eta$. Unless otherwise specified, we set the *flow ratio* to 0.5 and tune $\eta$ via grid search over $\{0.001, 0.01, 0.1, 0.3, 0.5\}$. We adopt the Adam optimizer with a learning rate of $3 \times 10^{-4}$. We ensure reliability by reporting OFQL results as the average D4RL normalized score (Fu et al., 2020), computed over three training seeds, with each model evaluated on 150 episodes per task. Other hyperparameters remain consistent with DQL. Additional implementation details are provided in Appendix B.

## 6 EXPERIMENTAL RESULT

Benchmark results are summarized in Table 1, with details discussed below.

**Locomotion Domain (MuJoCo).** OFQL achieves strong performance in locomotion tasks, surpassing competitive diffusion-based baselines such as DQL, DD . In particular, OFQL improves the average performance of DQL from 87.9 to 92.5, with notable gains on medium and medium-replay tasks, which are known to contain suboptimal and noisy trajectories. These tasks often induce complex, multi-modal action distributions that challenge standard policy learning, making expressive action modeling and stable value learning essential. The observed improvements may be attributed to two key aspects of OFQL: (1) its policy modeling remains expressive for capturing complex action distributions, and (2) the avoidance of BPTT in Q-learning, which may yield more stable value estimation, leading to better convergence. Together, these factors provide a plausible explanation for OFQL's consistent performance gains.

| Dataset | Non-Diffusion Policies | | | Diffusion Planners | | Multi-step Diffusion Policies | | | One-step Flow Policies | |
|---|---|---|---|---|---|---|---|---|---|---|
| | BC | TD3-BC | IQL | Diffuser | DD | EDP | IDQL | DQL | FQL | OFQL (Ours) |
| HalfCheetah-Medium-Expert | 55.2 | 90.7 | 86.7 | 90.3 ± 0.1 | 88.9 ± 1.9 | 95.8 ± 0.1 | 91.3 ± 0.6 | 96.8 ± 0.3 | **99.8** ± 0.1 | 95.2 ± 0.4 |
| Hopper-Medium-Expert | 52.5 | 98.0 | 91.5 | 107.2 ± 0.9 | 110.4 ± 0.6 | 110.8 ± 0.4 | 110.1 ± 0.7 | **111.1** ± 1.3 | 86.2 ± 1.3 | 110.2 ± 1.3 |
| Walker2d-Medium-Expert | 107.5 | 110.1 | 109.6 | 107.4 ± 0.1 | 108.4 ± 0.1 | 110.4 ± 0.0 | 110.6 ± 0.0 | 110.1 ± 0.3 | 100.5 ± 0.1 | **113.0 ± 0.1** |
| HalfCheetah-Medium | 42.6 | 48.3 | 47.4 | 43.8 ± 0.1 | 45.3 ± 0.3 | 50.8 ± 0.0 | 51.5 ± 0.1 | 51.1 ± 0.5 | 60.1 ± 0.1 | **63.8** ± 0.1 |
| Hopper-Medium | 52.9 | 59.3 | 66.3 | 89.5 ± 0.7 | 98.2 ± 0.1 | 72.6 ± 0.2 | 70.1 ± 2.0 | 90.5 ± 4.6 | 74.5 ± 0.2 | **103.6** ± 0.1 |
| Walker2d-Medium | 75.3 | 83.7 | 78.3 | 79.4 ± 1.0 | 79.6 ± 0.9 | 86.5 ± 0.2 | **88.1** ± 0.4 | 87.0 ± 0.9 | 72.7 ± 0.8 | 87.4 ± 0.1 |
| HalfCheetah-Medium-Replay | 36.6 | 44.6 | 44.2 | 36.0 ± 0.7 | 42.9 ± 0.1 | 44.9 ± 0.4 | 46.5 ± 0.3 | 47.8 ± 0.3 | 51.1 ± 0.1 | **51.2** ± 0.1 |
| Hopper-Medium-Replay | 18.1 | 60.9 | 94.7 | 91.8 ± 0.5 | 99.2 ± 0.2 | 83.0 ± 1.7 | 99.4 ± 0.1 | 101.3 ± 0.6 | 85.4 ± 0.5 | **101.9** ± 0.7 |
| Walker2d-Medium-Replay | 26.0 | 81.8 | 73.9 | 58.3 ± 1.8 | 75.6 ± 0.6 | 87.0 ± 2.6 | 89.1 ± 2.4 | 95.5 ± 1.5 | 82.1 ± 1.2 | **106.2** ± 0.6 |
| Average (MuJoCo) | 51.9 | 75.3 | 77.0 | 78.2 | 83.2 | 82.4 | 84.1 | 87.9 | 79.2 | **92.5** |
| AntMaze-Medium-Play | 0.0 | 10.6 | 71.2 | 6.7 ± 5.7 | 8.0 ± 4.3 | 73.3 ± 6.2 | 67.3 ± 5.7 | 76.6 ± 10.8 | 78.0 ± 7.0 | **88.1** ± 5.0 |
| AntMaze-Large-Play | 0.0 | 0.2 | 39.6 | 17.3 ± 1.9 | 0.0 ± 0.0 | 33.3 ± 1.9 | 48.7 ± 4.7 | 46.4 ± 8.3 | 84.0 ± 7.0 | **84.0** ± 6.1 |
| AntMaze-Medium-Diverse | 0.8 | 3.0 | 70.0 | 2.0 ± 1.6 | 4.0 ± 2.8 | 52.7 ± 1.9 | 83.3 ± 5.0 | 78.6 ± 10.3 | 71.0 ± 13.0 | **90.2** ± 4.2 |
| AntMaze-Large-Diverse | 0.0 | 0.0 | 47.5 | 27.3 ± 2.4 | 0.0 ± 0.0 | 41.3 ± 3.4 | 40.0 ± 11.4 | 56.6 ± 7.6 | **83.0** ± 4.0 | 76.1 ± 6.6 |
| Average (AntMaze) | 0.2 | 3.5 | 57.1 | 13.3 | 3.0 | 50.2 | 59.8 | 64.6 | 79.0 | **84.6** |
| Kitchen-Mixed | 51.5 | 0.0 | 51.0 | 52.5 ± 2.5 | **75.0 ± 0.0** | 50.2 ± 1.8 | 60.5 ± 4.1 | 62.6 ± 5.1 | 50.5±1.6 | 69.0 ± 1.5 |
| Kitchen-Partial | 38.0 | 0.0 | 46.3 | 55.7 ± 1.3 | 56.5 ± 5.8 | 40.8 ± 1.5 | **66.7** ± 2.5 | 60.5 ± 6.9 | 55.7±2.5 | 65.0 ± 2.3 |
| Average (Kitchen) | 44.8 | 0.0 | 48.7 | 54.1 | 65.8 | 45.5 | 66.6 | 61.6 | 53.1 | **67.0** |

Table 1: Comparison of normalized scores on D4RL benchmark across MuJoCo, Kitchen, and AntMaze domains. Bold values indicate the best performance per row.

Compared to other acceleration approaches, although EDP and IDQL enhance efficiency and stability, they do so at the expense of reducing final performance relative to DQL, whereas OFQL achieves improvements. When compared with one-step FQL (based on distillation), OFQL surpasses it by a significant margin (+13.3). Overall, OFQL offers a superior combination of efficiency and effectiveness, with consistent gains across varying data regimes.

**AntMaze Domain.** AntMaze tasks are particularly challenging due to sparse rewards and suboptimal demonstrations, requiring accurate and stable Q-value guidance to perform well. Prior approaches (e.g., BC and Diffuser) struggle without Q-learning, whereas methods incorporating Q-learning signals (e.g., IDQL and DQL) achieve consistently better results.

Building on the Q-learning framework, while EDP and IDQL underperform relative to DQL, OFQL achieves a substantial improvement over DQL, raising performance from 64.6 to 84.6 and outperforming all diffusion-based baselines and the one-step FQL. Notably, FQL, which employs a one-step policy for actor–critic training, improves upon DQL from 64.6 to 79.0. We assume Q-learning is crucial in this domain, and that OFQL and FQL may benefit from avoiding BPTT.

**Kitchen Domain.** . The Kitchen datasets contain low-entropy, narrowly distributed behaviors (Dong et al., 2024), where action modeling plays a larger role than Q-learning. OFQL surpasses DQL (61.6 → 67.0), achieving the strongest performance across methods. Although IDQL proves competitive, FQL drops to 53.1, likely because its one-step policy lacks sufficient expressivity. Remarkably, OFQL's one-step formulation does not suffer this drawback, instead preserving expressivity while achieving state-of-the-art results.

# 7 ABLATION STUDY

| Method (Steps) | DQL (5) | DQL+DDIM (1) | FBRAC (1) | FQL (1) | OFQL (1) |
|---|---|---|---|---|---|
| **Score** | 87.9 | 11.6 (-76.3) | 67.1 (-20.8) | 79.2 (-8.7) | 92.6 (+4.7) |

Table 2: Comparison of methods (steps) using different improvement strategies toward one-step action generation across 9 MuJoCo tasks. The average normalized score is reported.

**Comparison of Strategies Toward One-Step Prediction.** To investigate how to effectively adapt DQL for one-step prediction, we evaluate the following strategies: (1) **DQL:** The base model, trained and evaluated with 5 denoising steps. (2) **DQL+DDIM:** A pretrained DQL model with a one-step DDIM sampler applied only at inference time. (3) **FBRAC:** The flow policy-based counterpart of DQL, trained with 5 denoising steps for actor–critic updates but using a single step at inference. (4) **FQL:** Learns a behavioral policy with a flow model, then distills it into a one-step policy for actor–critic training and inference. (5) **OFQL:** Trained and evaluated entirely in the one-step regime.

Table 2 reports the average performance across 9 MuJoCo tasks. DQL+DDIM shows severe degradation ($-76.3$), suggesting that direct application of improved samplers for one-step inference is ineffective. FBRAC performs better ($-20.8$), but still lags behind DQL, likely due to the discretization error introduced when performing one-step prediction with curved trajectories. FQL further narrows the gap ($-8.7$) by distilling a one-step policy from a multi-step flow model. In contrast, OFQL achieves the best results, exceeding DQL by ($+4.7$) while consistently supporting one-step sampling in both training and inference.

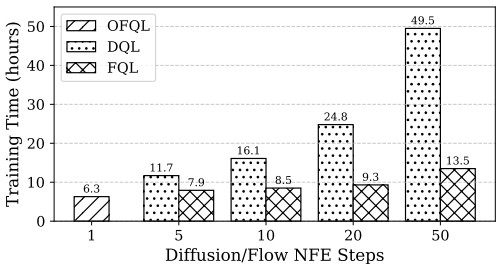
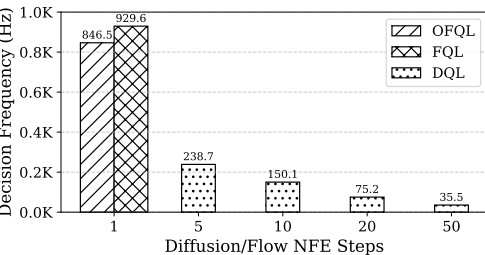

Figure 3: Training Time ($\downarrow$) and Decision Frequency ($\uparrow$) over one million steps, averaged on MuJoCo tasks. NFE (Number of Function Evaluations) denotes the denoising steps required by a flow/diffusion model to generate one action from pure noise. During training and inference, OFQL uses only one NFE, while DQL requires multiple ones. It is worth noting that for inference, FQL runs with a one-step policy, but training still relies on a multi-step flow policy to construct distillation targets.

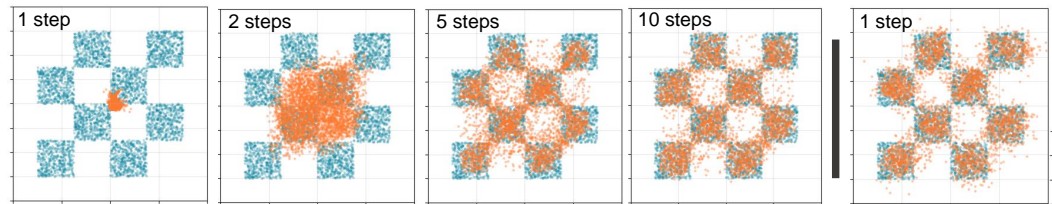

Figure 4: Comparison of distribution modeling capabilities between FM with marginal velocity parameterization (left; evaluated at 1,2,5,10 steps generation) and average velocity parameterization (right; evaluated with one-step generation) on a toy dataset with complex multi-modal structure.

**Training and Inference Efficiency Comparison.** Figure 3 reports the wall-clock training time (1M steps) and decision frequency (Lu et al., 2025b) on an A100 GPU (see Appendix D for experimental protocol). DQL's training time scales nearly linearly with the number of denoising steps—from 11.7 hours at 5 steps to 49.5 hours at 50 steps—while OFQL completes training in only 6.3 hours. At inference, OFQL reaches 846.5 Hz, compared to 238.7 Hz for 5-step DQL and just 35.5 Hz for 50-step DQL.

Compared to a one-step FQL baseline, OFQL achieves nearly the same decision frequency but enjoys shorter training time. This advantage arises because FQL requires multiple NFEs to compute distillation targets, leading to a slower training loop. Note that, despite comparable speed, FQL consistently underperforms OFQL in terms of policy performance.

Overall, OFQL achieves substantially faster training and higher decision frequency without sacrificing model expressivity, making it more practical than multi-step DQL or distillation-based FQL.

**Ablation on flow ratio.** We study the effect of varying the flow ratio across different datasets in HalfCheetah (Table 3). The best performance is obtained at a flow ratio of 0.5, achieving 95.2 on Medium Expert, 63.8 on Medium, and 52.2 on Medium Replay. In contrast, using either the flow ratio equal to 1 (equivalent to pure flow matching) or setting it to 0 results in noticeable performance degradation. A moderate flow ratio serves as an effective regularizer, yielding the most stable and robust learning behavior.

| Flow Ratio | 1 | 0.75 | 0.5 | 0.25 | 0 |
|---|---|---|---|---|---|
| Medium Expert | 38.3 | 90.86 | **95.2** | 92.03 | 90.47 |
| Medium | 46.3 | 62.03 | **63.8** | 63.76 | 63.2 |
| Medium Replay | 45.2 | 50.2 | **51.2** | 50.3 | 10.5 |

Table 3: D4RL scores across HalfCheetah datasets under varying *flow ratios*.

**Compare Marginal Velocity and Average Velocity Parameterization.** DQL has convincingly shown that employing a more expressive policy leads to superior final performance in the actor-critic training framework. To examine the expressiveness of one-step generation, we conduct a toy dataset experiment comparing Flow Matching with marginal velocity ($v$-param) versus average velocity ($u$-param) parameterization across different generation steps.. As illustrated in the rightmost panel of Figure 4, samples generated by $u$-param in a single step already demonstrate strong mode coverage and close alignment with the target distribution. In contrast, $v$-param requires multiple steps to achieve comparable quality and often produces collapsed samples with fewer steps. These results underscore the advantage of modeling the average velocity field for one-step generation and give strong confidence to modeling policy. Additional experimental results and experiment setting are provided in the Appendix C.

## 8 CONCLUSION

We presented One-Step Flow Q-Learning (OFQL), a novel policy learning framework that overcomes key limitations of Diffusion Q-Learning by enabling efficient, single-step action generation during both training and inference. By reformulating DQL within the Flow Matching framework and learning an average velocity field rather than a marginal one, OFQL eliminates the need for multi-step denoising, recursive gradient propagation. This leads to faster training and inference, while surpassing the performance of state-of-the-art diffusion-based offline RL methods. Empirical results on the D4RL benchmark confirm the effectiveness and efficiency of OFQL, underscoring the promise of one-step flow policies for advancing offline RL. More broadly, OFQL facilitates accurate high-frequency decision-making, suggesting potential for real-time control and scalable deployment in complex, latency-sensitive domains.

## 9 ACKNOWLEDGMENTS

This work was supported by the Institute for Information & communications Technology Planning & Evaluation (IITP) grant funded by the Korea government (MSIT) (No. RS-2022-II0951, Development of Uncertainty-Aware Agents Learning by Asking Questions) and the National Research Foundation of Korea (NRF) grant funded by the Korea government (MSIT) (RS-2025-24742969, Intelligent Robotic System using Continual Learning and Multimodal Language Model based Multi Attribute Feedback).

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

## A  BENCHMARKING TASKS AND EVALUATION PROTOCOL

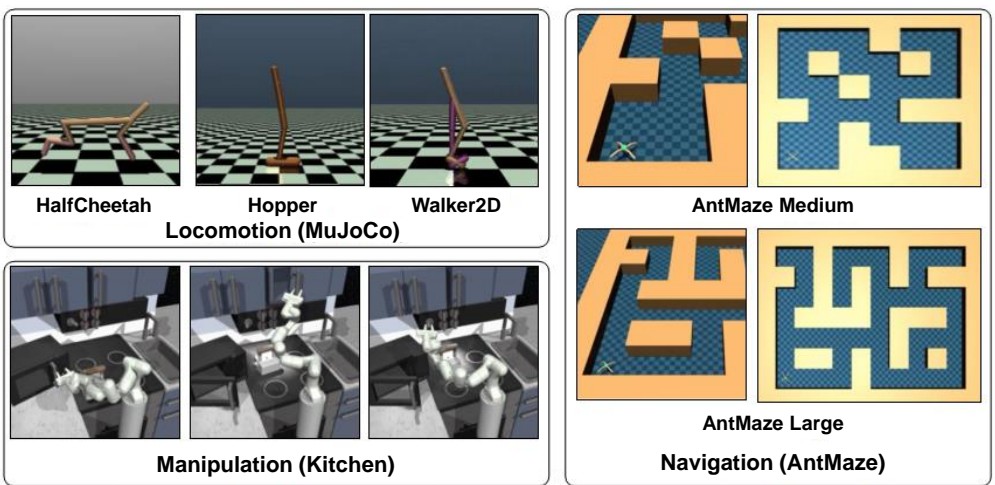

Figure 5: Illustration of the benchmarking tasks examined in this study. The tasks include locomotion challenges for short-term decision-making, robotic arm manipulation tasks requiring long-term strategic decision-making, and navigation tasks focused on path optimization.

**Benchmarking tasks.** As shown in Figure 5, we evaluate the performance of OFQL using a diverse set of benchmarking tasks that span various domains of reinforcement learning. These tasks are chosen to assess OFQL's capability across a broad spectrum of environment setups, which is essential for understanding the model's robustness and generalization. The selected tasks include locomotion challenges that emphasize short-term decision-making, robotic arm manipulation tasks requiring long-term strategic decision-making, and navigation tasks focused on pathfinding. By covering this wide array of tasks, we ensure a comprehensive evaluation of OFQL's performance in both simple and complex settings, facilitating a deeper understanding of its strengths and limitations.

Locomotion (MuJoCo): The MuJoCo Locomotion task is a well-established benchmark in reinforcement learning, where the agent is tasked with controlling a simulated robot to navigate through a dynamic and complex environment. This task is designed to test the agent's ability to perform locomotion tasks, emphasizing short-term decision-making and agility in navigating unpredictable terrains.

Manipulation (Kitchen): The Kitchen (Franka Kitchen) task is a robotic arm manipulation challenge in which the agent is required to interact with objects in a kitchen environment. This task is specifically designed to evaluate the agent's proficiency in long-term strategic decision-making, as it involves making sequences of actions for tasks such as object manipulation and coordination, which require higher levels of temporal reasoning.

Navigation (AntMaze): The AntMaze task combines locomotion and planning challenges in a maze environment, where the agent must navigate through increasingly complex and variable maze configurations. This task is designed to test the agent's ability to perform locomotion tasks while incorporating advanced planning strategies, balancing exploration and exploitation in a maze with dynamic elements.

**Evaluation Metric.** We adopt the D4RL (Fu et al., 2020) benchmark to report the normalized score, which allows for fair comparison across approaches. The *normalized score* is computed for each environment, using the following formula:

$$\text{Normalized Score} = 100 \times \frac{\text{score} - \text{random score}}{\text{expert score} - \text{random score}} \tag{18}$$

A normalized score of 0 represents the average returns (over 100 episodes) of an agent that selects actions uniformly at random across the action space. A normalized score of 100 corresponds to the average returns of a domain-specific expert (chosen by D4RL).

## B ARCHITECTURAL AND IMPLEMENTATION DETAILS

---

**Algorithm 1** OFQL Algorithm

---

1: **Initialize** policy network $\pi_\theta$, $\pi_{\theta'}$, critic networks $Q_{\phi_1}$ and $Q_{\phi_2}$, $Q_{\phi_1'}$, $Q_{\phi_2'}$
2: **for** each iteration **do**
3:     Sample transition mini-batch $\mathcal{B} = \{(s_h, a_h, r_h, s_{h+1})\} \sim \mathcal{D}$
4:     **# Q-value function learning**
5:     Sample $a_{h+1} \sim \pi_{\theta'}(a_{h+1} \mid s_{h+1})$ by Eq. 16
6:     Update $Q_{\phi_1}$ and $Q_{\phi_2}$ using Eq. 6 {Max-Q backup (Kumar et al., 2020) optional}
7:     **# Policy learning**
8:     Sample $a_h \sim \pi_\theta(a_h \mid s_h)$ by Eq. 16
9:     Update policy $\pi_\theta$ by minimizing Eq. 7
10:    **# Update target networks every K iteration**
11:    $\theta' \leftarrow \rho\theta' + (1 - \rho)\theta$
12:    $\phi_i' \leftarrow \rho\phi_i' + (1 - \rho)\phi_i$ for $i = \{1, 2\}$
13: **end for**

---

Our approach generally builds directly on DQL (Wang et al., 2022), inheriting its training and inference. Below, we outline the key architectural and implementation details.

**Architectural Details.** We adopt the original DQL architecture for both the Q-function and policy networks, with a minor modification to the policy input. Specifically, we augment the input by concatenating an additional positional embedding corresponding to the target step $r$, alongside the standard timestep embedding $t$. More specifically, the architectures are as below:

Policy Network: The policy is modeled as the average velocity function $u_\theta(a_t, r, t; s)$, where $a_t$ denotes the action latent, $t$ and $r$ are timestep variables, and $s$ is the state conditioning input. We adopt the same MLP-based architecture as used in DQL, with the modification of incorporating the additional timestep $r$. Specifically, $u_\theta$ is parameterized as a 3-layer multilayer perceptron (MLP) with Mish activations and 256 hidden units per layer, followed by a linear output layer that maps to the action dimension. The input to $u_\theta$ is the concatenation of the action latent vector, the current state vector, and the sinusoidal positional embeddings of timesteps $t$ and $r$ (time embedding size 64). The output is the predicted average velocity that flows from timestep $t$ to $r$.

Q Networks: We utilize the same Q network architecture as in DQL. Specifically, we employ two Q networks, each implemented as a 3-layer MLP with Mish activations and 256 hidden units per layer, followed by a linear output layer that maps to a single action-value dimension. The input to each Q network is the concatenation of the action and the observation, and the output is the estimated state-action value.

**Training Details.** The pseudo algorithm of OFQL is provided in Algorithm 1, where Max-Q backup is applied to AntMaze tasks only, as in DQL. In the training, the time variables $t$ and $r$ are sampled from a logit-normal distribution (Esser et al., 2024) with parameters $(-0.4, 1.0)$, which improves stability compared to uniform sampling. During sampling, time pairs are selected such that $r \neq t$ holds for 50% of the samples (i.e., *flow ratio* equal to 0.5). In the actor loss, the hyperparameter $\alpha$ balances behavior regularization and Q value maximization. To normalize for dataset-specific Q-value scales, $\alpha$ is adapted as $\alpha = \frac{\eta}{\mathbb{E}_{(s,a)\sim\mathcal{D}}\left[\|Q_\phi(s,a)\|\right]}$, where $\eta$ is a tunable hyperparameter. We search $\eta$ over $\{0.001, 0.01, 0.1, 0.3, 0.5\}$ since the relative importance of Q-guidance varies by domain (e.g., the Kitchen tasks require more policy regularization and the AntMaze tasks require more Q-learning).

Training is conducted for 1000 epochs (2000 for MuJoCo tasks), with each epoch consisting of 1000 gradient steps and a batch size of 256. Both the policy and Q networks are optimized with Adam (Kingma & Ba, 2014), using a learning rate of $3 \times 10^{-4}$. For rewards, we adopt the original task rewards in MuJoCo Gym and Kitchen, while following CQL's modification (Kumar et al., 2020) for AntMaze, consistent with DQL. For evaluation, we follow the protocol of Dong et al. (2024),

including the same sampling procedure. We report the mean normalized return averaged over three training seeds, with each model evaluated on 150 episodes per task.

## C DDPM, FLOW MATCHING WITH MARGINAL VELOCITY COMPARED TO AVERAGE VELOCITY PARAMETERIZATION ON TOY DATASETS

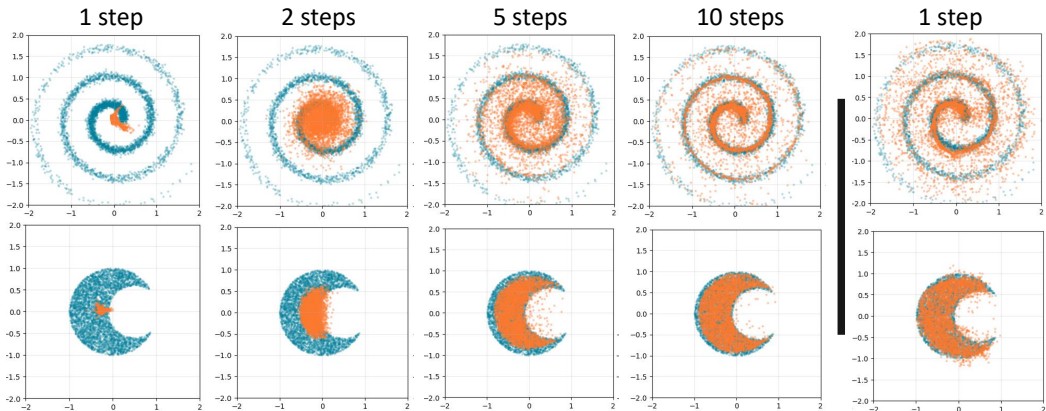

Figure 6: Comparison of sample quality between v-parameterization (left, steps = 1, 2, 5, 10) and u-parameterization (right, one step) on toy datasets.

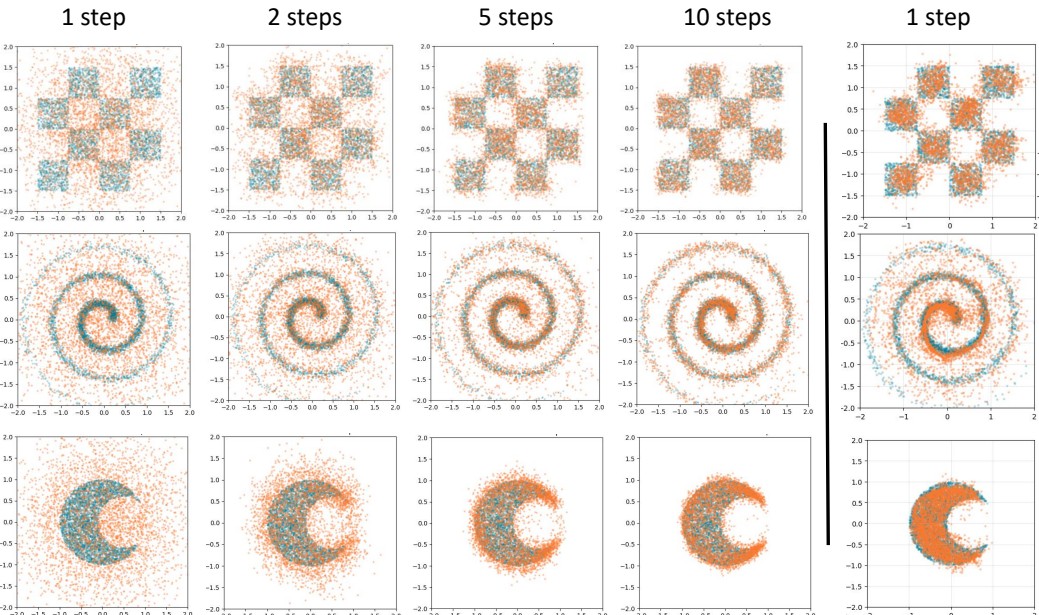

Figure 7: Comparison of sample quality between DDPM in DQL (left, steps = 1, 2, 5, 10) and u-parameterization in OFQL (right, one step) on toy datasets.

**Experiment Setup.** As DQL convincingly demonstrates, greater model expressiveness leads to stronger final performance in the actor–critic framework. To illustrate the capability of modeling complex distributions in a one-step setting, we compare (1) Flow Matching with two velocity parameterizations; marginal velocity ($v$-param) and average velocity ($u$-param) (2) DDPM in DQL with average velocity in OFQL; across three synthetic datasets: Crescent, Spiral, and Checkerboard. Each dataset challenges the models to capture distinct geometric structures.

1. Crescent: Points form a crescent shape, with an outer and inner circle, testing the models' ability to avoid points in the inner circle.

2. Spiral: Points follow a noisy spiral, evaluating the models' robustness in recovering the structure amidst noise.

3. Checkerboard: Points are arranged in a checkerboard pattern, testing the models' capacity to capture multimodal distributions.

**Architecture.** (1) $v$-param: estimates marginal velocity using a multi-layer perceptron (MLP) with 3 hidden layers (64 units each), with inputs: noise latent, timestep. (2) $u$-param: estimates average velocity using a similar MLP architecture but with an additional target time input. (3) DDPM: estimate the time step noise using a multi-layer perceptron (MLP) with 3 hidden layers (64 units each), with inputs: noise latent, timestep. All models use Mish activations.

**Training and Evaluation.** We train both models for 100 epochs with a batch size of 2048 and 40 batches for each epoch. The DDPM, $v$-param is evaluated with varying prediction steps (1, 2, 5, 10), while the $u$-param is evaluated with one denoising step. We visualize the ground truth distribution (blue points) and the generated samples (orange points) on a 2D plot to assess how well each model captures the dataset's geometric structure.

**Results on $v$-param vs $u$-param** While the main paper reports results on the Checkerboard dataset, Figure 6 additionally presents results on two other toy datasets: Spiral and Moon. The results show that the $u$-param consistently generates well-structured samples that match the target distribution even in a single step. In contrast, the $v$-param exhibits significant mode collapse and noise in early steps (1–2), requiring up to 10 steps to approximate the target shape. These findings reinforce our main conclusion: modeling the average velocity field enables the $u$-param to achieve accurate and efficient one-step generation, outperforming standard flow models across various geometrically complex distributions.

**Results on DDPM vs $u$-param.** Figure 7 illustrates that the one-step $u$-parameterization in OFQL achieves expressivity comparable to a full DDPM, despite requiring only a single forward pass. In contrast, DDPM produces highly noisy outputs when restricted to a small number of steps (1–2) and typically requires around 10 denoising steps to approximate the target distribution reliably. Importantly, one-step generation is particularly advantageous in actor–critic RL, as it eliminates the need for backpropagation through time. This makes the one-step $u$-parameterization a more practical choice than DDPM for policy learning, even though both approaches exhibit similar expressive capacity.

## D    TRAINING AND INFERENCE EFFICIENCY COMPARISON

We evaluate the decision frequency and training time of our method and baseline across 9 Mujoco tasks. The decision frequency (Lu et al., 2025b) reflects the number of actions (or action batches) generated per second by the evaluated model.

Experiments are conducted on an Ubuntu server with an Intel(R) Xeon(R) Gold 5317 CPU @ 3.00GHz (48 cores, 96 threads) and an A100 PCIe 80 GB GPU. The wall-clock training time (in hours) is measured over 1 million training steps and averaged across the 9 Mujoco tasks. Decision frequency is measured over 3000 action batches (batch size 2500) for each task and averaged across all tasks.

## E    LIMITATIONS AND FUTURE WORK

The efficiency and expressivity of OFQL position it as a compelling approach for advancing real-world reinforcement learning, in conjunction with existing approaches (Venkataraman et al., 2025; Nguyen et al., 2021a; Luu et al., 2024; 2025b;a; Nguyen et al., 2024b;a). In particular, OFQL enables one-step action generation while achieving state-of-the-art performance, thereby supporting decision-making at frequencies suitable for real-time robotics, autonomous driving, medical and related domains (Ramstedt & Pal, 2019; Kiran et al., 2021; Ouyang et al., 2022; Yoon et al., 2025; Pham et al., 2022). Its reduced training and inference cost also lowers the computational barrier for

scaling reinforcement learning to larger datasets and more complex tasks, providing a practical path toward widespread industrial adoption.

While OFQL demonstrates compelling efficiency and performance gains in offline reinforcement learning, our current evaluation focuses primarily on single-goal state-based decision-making tasks—specifically those relying on low-dimensional proprioceptive observations from the D4RL benchmark. This leaves several promising directions for future exploration.

First, extending OFQL to online reinforcement learning presents a natural next step. Our one-step formulation removes the computational bottlenecks that typically hinder real-time interaction, making OFQL a promising candidate for scalable online learning. Investigating stability and sample-efficiency in this setting remains an important open question.

Second, we aim to generalize OFQL to vision-based or point-cloud-based control settings, where observations are high-dimensional. An open question is whether we can leverage encoder components from domain-specific architectures for feature extraction (e.g., (He et al., 2016; Zhou & Tuzel, 2018; Vu et al., 2022; 2023; 2021)) or whether a specialized encoder tailored to OFQL should be designed. Figuring out effective architectures and integrating them with one-step flow-based policies could pave the way for end-to-end learning in more complex and unstructured environments.

Third, future work could explore extending OFQL to goal-conditioned and multi-task RL settings. Learning conditional average velocity fields to support diverse goal-directed behaviors—without resorting to separate diffusion or reward models—would offer greater flexibility and generalization.

Overall, OFQL provides a general foundation for fast and expressive policy learning, and we hope future work expands its applicability across broader domains and learning paradigms.

## F  DETAILED ON DIFFUSION MODELS, FLOW MATCHING

For completeness, we provide background on diffusion models and flow matching, which serve as the foundations for our method.

**Diffusion Models.** Diffusion models are a high-performing class of generative models that learn to sample from an unknown data distribution $q(x^0)$ using a dataset drawn from it (Ho et al., 2020; Song & Ermon, 2019; Song et al., 2020b; Sohl-Dickstein et al., 2015). Denoising Diffusion Probabilistic Models (DDPMs) (Ho et al., 2020), the canonical diffusion model used in DQL, define a forward diffusion process $q(x^{1:K} \mid x^0) = \prod_{k=1}^{K} q(x^k \mid x^{k-1})$ as a fixed Markov chain that gradually corrupts data with Gaussian noise over $K$ steps, where $q(x^k \mid x^{k-1}) = \mathcal{N}\left(x^k; \sqrt{1 - \beta^k}x^{k-1}, \beta^k \boldsymbol{I}\right)$, and the variance schedule $\{\beta^k\}_{k=1}^{K}$ is predefined such that as $K \to \infty$, $x^K$ approaches an isotropic Gaussian.

The corresponding reverse process, enables generating data from pure noise, is parameterized by $\psi$ and defined as:

$$p_\psi(x^{0:K}) = \mathcal{N}(x^K; \boldsymbol{0}, \boldsymbol{I}) \prod_{k=1}^{K} p_\psi(x^{k-1} \mid x^k), \tag{19}$$

which is learned by maximizing the variational lower bound $\mathbb{E}_q \left[ \log \frac{p_\psi(x^{0:K})}{q(x^{1:K}|x^0)} \right]$ (Blei et al., 2017; Ho et al., 2020).

After training, sampling from $q(x^0)$ is approximated by drawing $x^K \sim \mathcal{N}(\boldsymbol{0}, \boldsymbol{I})$ and applying the reverse Markov chain from $k = K$ to $k = 1$ via the learn model $p_\psi$. Conditional generation is straightforwardly supported via $p_\psi(x^{k-1} \mid x^k, c)$.

**Flow Matching.** Flow Matching (FM) (Lipman et al., 2022) is a generative modeling framework that learns deterministic velocity fields to directly transport noise to data along smooth, stable trajectories.

Given data $x \sim q(x)$ and noise $\epsilon \sim p_{\text{prior}}(\epsilon)$, FM defines a linear flow path:

$$z_t = \alpha_t x + \beta_t \epsilon, \quad v_t = \frac{dz_t}{dt} = \dot{\alpha}_t x + \dot{\beta}_t \epsilon, \tag{20}$$

where $\alpha_t, \beta_t$ are predefined schedules (e.g., $\alpha_t = 1 - t$, $\beta_t = t$), and the dot notation (e.g., $\dot{\alpha}_t$) denotes the time derivative with respect to the continuous flow step $t \in [0, 1]$. The *conditional*

*velocity* $v_t(z_t \mid x)$ captures the direction of flow for a specific sample, and the *marginal velocity* field is defined as the expectation:

$$v(z_t, t) \triangleq \mathbb{E}_{p_t(v_t \mid z_t)}[v_t]. \tag{21}$$

FM essentially models the marginal velocity, as it is feasible to approximate this field; parametrized by the neural network $v_\theta(z_t, t)$;using the Conditional Flow Matching loss:

$$\mathcal{L}_{\mathrm{CFM}}(\theta) = \mathbb{E}_{t, x, \epsilon} \|v_\theta(z_t, t) - v_t(z_t \mid x)\|^2, \tag{22}$$

where, under the commonly used schedule $\alpha_t = 1 - t$, $\beta_t = t$, the conditional velocity simplifies to $v_t(z_t \mid x) = \epsilon - x$.

In inference, sampling is performed by solving the ODE in reverse time:

$$\frac{dz_t}{dt} = v(z_t, t), \quad \text{starting from } z_1 = \epsilon \sim p_{\mathrm{prior}}(\epsilon), \tag{23}$$

where the solution is approximated using a numerical solver, such as Euler's method: $z_{t-\Delta t} = z_t - \Delta t \cdot v(z_t, t)$.

**On the classifier-free guidance** While classifier-free guidance (CFG) might be considered to better align generated samples with the conditioning variable $c$ in the image generation domain, prior work has shown that CFG can lead to undesirable behaviors in sequential decision-making tasks. Specifically, CFG tends to bias the generation process toward high-density regions associated with $c$, which may cause agents to overlook high-return trajectories critical for long-horizon planning (Pearce et al., 2023). Additionally, DQL adopts a no-guidance approach. For a fair comparison, we follow the design choices made in DQL and prior works (Chi et al., 2023; Wang et al., 2022) and adopt a *no-guidance* paradigm, ensuring stable and unbiased policy generation.

## G   FORMAL JUSTIFICATION OF ACTION ACCURACY PRESERVATION IN ONE-STEP GENERATION THROUGH AVERAGE VELOCITY FIELD

We show that under perfect learning (no estimation error), learning the conditional average velocity $u_\theta(a_t, r, t; s)$ and applying the one-step update in Eq. 12 recovers the same endpoint map as integrating the conditional Flow Matching dynamics, thereby enabling accurate one-step action generation.

Let $v^\star(a, t; s)$ be the ground-truth conditional marginal velocity field assumed to govern the Flow Matching (FM) dynamics that generate the target action distribution $\mu(\cdot \mid s)$. In FM, this velocity field transports Gaussian noise to data through the ODE

$$\frac{da_t}{dt} = v^\star(a_t, t; s), \qquad a_1 = \epsilon \sim \mathcal{N}(0, I). \tag{24}$$

Solving this ODE backward from $t = 1$ to $t = 0$ yields an endpoint $a_0$ that depends on both the noise sample $\epsilon$ and the conditioning state $s$. This defines the FM endpoint map

$$T^\star(\epsilon, s) = a_0 = \epsilon - \int_0^1 v^\star(a_\tau, \tau; s) \, d\tau. \tag{25}$$

The push-forward of this map over the Gaussian prior, $(T^\star)_\# \mathcal{N}(0, I)$, recovers the target action distribution $\mu(\cdot \mid s)$.

For any interval $[r, t] \subseteq [0, 1]$, define the average velocity as

$$u^\star(a_t, r, t; s) = \frac{1}{t - r} \int_r^t v^\star(a_\tau, \tau; s) \, d\tau, \tag{26}$$

which represents the net displacement of the FM trajectory over this interval. Applying this definition to $[0, 1]$ gives

$$u^\star(a_1, 0, 1; s) = u^\star(\epsilon, 0, 1; s) = \int_0^1 v^\star(a_\tau, \tau; s) \, d\tau, \tag{27}$$

and therefore the endpoint map satisfies

$$T^\star(\epsilon, s) = \epsilon - \int_0^1 v^\star(a_\tau, \tau; s)\, d\tau = \epsilon - u^\star(\epsilon, 0, 1; s).$$

Suppose a learned model $u_\theta$ (e.g., a model by a neural network) approximates this average velocity $u^\star$ on the support of the Gaussian prior without estimation error. In that case, the one-step generator in Eq. 12 becomes:

$$T_\theta(\epsilon, s) = \epsilon - u_\theta(\epsilon, 0, 1; s) = \epsilon - u^\star(\epsilon, 0, 1; s) = T^\star(\epsilon, s). \tag{28}$$

Because both maps push the Gaussian prior forward in the same way, their induced action distributions coincide:

$$(T_\theta)_\# \mathcal{N}(0, I) = (T^\star)_\# \mathcal{N}(0, I) = \mu(\cdot \mid s).$$

Hence, the learned policy

$$\pi_\theta(a \mid s) \triangleq (T_\theta)_\# \mathcal{N}(0, I)$$

matches the target distribution exactly, demonstrating that the learned average velocity preserves the FM action accuracy in a single forward pass without ODE integration.

# H  FORMAL JUSTIFICATION OF AVERAGE-VELOCITY LEARNING ENCOURAGES THE LEARNED ONE-STEP POLICY TO STAY CLOSE TO THE BEHAVIOR POLICY

We show that, under general (imperfect) learning conditions, minimizing the average-velocity matching loss $\mathcal{L}_{\text{FBC}^\star}(\theta)$ ensures that the learned one-step policy $\pi_\theta(\cdot \mid s)$ converges toward the behavior distribution $\mu(\cdot \mid s)$. In particular, $\mathcal{L}_{\text{FBC}^\star}(\theta)$ upper-bounds the squared 2-Wasserstein distance between $\pi\theta(\cdot \mid s)$ and $\mu(\cdot \mid s)$ - up to a positive constant—implying that small average-velocity error enforces closeness between the two distributions.

Let $\epsilon \sim \mathcal{N}(0, I_d)$ be a $d$-dimensional standard Gaussian. For each state $s \in \mathcal{S}$, define

$$T^\star(\epsilon, s) = \epsilon - \int_0^1 v^\star(a_\tau, \tau; s)\, d\tau = \epsilon - u^\star(\epsilon, 0, 1; s), \qquad T_\theta(\epsilon, s) = \epsilon - u_\theta(\epsilon, 0, 1; s),$$

so that the induced action distributions are the push-forwards

$$\mu(\cdot \mid s) = (T^\star(\cdot, s))_\# \mathcal{N}(0, I), \qquad \pi_\theta(\cdot \mid s) = (T_\theta(\cdot, s))_\# \mathcal{N}(0, I).$$

Recall the average-velocity matching loss:

$$\mathcal{L}_{\text{FBC}^\star}(\theta) = \mathbb{E}_{0 \le r \le t \le 1;\, s;\, \epsilon}\left[\left\|u_\theta(a_t, r, t; s) - u^\star(a_t, r, t; s)\right\|_2^2\right], \tag{29}$$

where $a_t$ is the (deterministic) solution of the flow-matching ODE at time $t$ given the initial noise $\epsilon$ and state $s$.

Assume that the sampling distribution over time pairs $(r, t)$ assigns a non-zero probability $p_{01} > 0$ to the endpoint pair $(0, 1)$, i.e. $\mathbb{P}[(r, t) = (0, 1)] = p_{01} > 0$. Then

$$\mathcal{L}_{\text{FBC}^\star}(\theta) = \mathbb{E}_{(r,t);\, s;\, \epsilon}\left[\left\|u_\theta(a_t, r, t; s) - u^\star(a_t, r, t; s)\right\|_2^2\right] \tag{30}$$

$$\ge p_{01} \mathbb{E}_{(r,t)=(0,1);\, s;\, \epsilon}\left[\left\|u_\theta(a_t, r, t; s) - u^\star(a_t, r, t; s)\right\|_2^2\right] \tag{31}$$

$$= p_{01} \mathbb{E}_{s;\, \epsilon}\left[\left\|u_\theta(\epsilon, 0, 1; s) - u^\star(\epsilon, 0, 1; s)\right\|_2^2\right]. \tag{32}$$

Using the endpoint parameterization

$$T_\theta(\epsilon, s) = \epsilon - u_\theta(\epsilon, 0, 1; s), \qquad T^\star(\epsilon, s) = \epsilon - u^\star(\epsilon, 0, 1; s),$$

we obtain the identity

$$\|u_\theta(\epsilon, 0, 1; s) - u^\star(\epsilon, 0, 1; s)\|_2^2 = \|T_\theta(\epsilon, s) - T^\star(\epsilon, s)\|_2^2.$$

Thus

$$\mathcal{L}_{\text{FBC}^\star}(\theta) \geq p_{01}\mathbb{E}_{s;\,\epsilon\sim\mathcal{N}(0,I)}\left[\|T_\theta(\epsilon,s) - T^\star(\epsilon,s)\|_2^2\right]. \tag{33}$$

For each state $s$, let $\lambda_s$ denote the joint distribution of $(T_\theta(\epsilon,s), T^\star(\epsilon,s))$ induced by $\epsilon \sim \mathcal{N}(0,I)$. Then $\lambda_s$ is a valid coupling between $\pi_\theta(\cdot \mid s)$ and $\mu(\cdot \mid s)$, i.e. $\lambda_s \in \Lambda(\pi_\theta(\cdot \mid s), \mu(\cdot \mid s))$. Therefore,

$$\mathbb{E}_\epsilon\left[\|T_\theta(\epsilon,s) - T^\star(\epsilon,s)\|_2^2\right] = \mathbb{E}_{(a,a^*)\sim\lambda_s}\left[\|a - a^*\|_2^2\right] \tag{34}$$

$$\geq \inf_{\lambda\in\Lambda(\pi_\theta(\cdot|s),\mu(\cdot|s))} \mathbb{E}_{(a,a^*)\sim\lambda}\left[\|a - a^*\|_2^2\right] \tag{35}$$

$$= W_2^2\big(\pi_\theta(\cdot \mid s),\,\mu(\cdot \mid s)\big), \tag{36}$$

where $W_2$ denotes the 2-Wasserstein distance on the action space with Euclidean ground metric. Combining the inequalities yields

$$\mathcal{L}_{\text{FBC}^\star}(\theta) \geq p_{01}\,\mathbb{E}_s\left[W_2^2\big(\pi_\theta(\cdot \mid s), \mu(\cdot \mid s)\big)\right]. \tag{37}$$

Thus, up to the positive constant factor, $\mathcal{L}_{\text{FBC}^\star}(\theta)$ upper-bounds the expected squared 2-Wasserstein distance between the learned policy $\pi_\theta(\cdot \mid s)$ and the target policy $\mu(\cdot \mid s)$ induced by flow matching. In particular, if $\mathcal{L}_{\text{FBC}^\star}(\theta) \to 0$, then $\mathbb{E}_s\left[W_2^2\big(\pi_\theta(\cdot \mid s), \mu(\cdot \mid s)\big)\right] \to 0$. Consequently, average-velocity learning regularizes $\pi_\theta(\cdot \mid s)$ toward the behavior distribution $\mu(\cdot \mid s)$, while still allowing complex, multimodal action distributions via the nonlinear endpoint map induced by flow-matching dynamics.

# I GRADIENT ANALYSIS OF OFQL ACTOR LOSS

The OFQL actor minimizes

$$\arg\min_\theta \mathcal{L}(\theta) = \arg\min_\theta \left(\mathcal{L}_{\text{FBC}}(\theta) - \alpha\,\mathcal{L}_{\text{Q}}(\theta)\right), \tag{38}$$

$$\mathcal{L}_{\text{Q}}(\theta) \triangleq \mathbb{E}_{s\sim\mathcal{D},\,a\sim\pi_\theta}\left[Q_\phi(s,a)\right]. \tag{39}$$

The OFQL actor loss jointly (i) maximizes the critic value (i.e., return) and (ii) keeps the policy close to the behavior distribution via FBC (see formal justification in H )

We now expand the gradient of each term.

**Gradient of the Q-term.** Recall that actions are sampled in one step:

$$a = \epsilon - u_\theta(\epsilon, 0, 1; s), \qquad \epsilon \sim \mathcal{N}(0,I), \tag{40}$$

The Q-term becomes

$$\mathcal{L}_{\text{Q}}(\theta) = \mathbb{E}_{s,\epsilon}\left[Q_\phi\big(s, \epsilon - u_\theta(\epsilon, 0, 1; s)\big)\right]. \tag{41}$$

Applying the chain rule:

$$\nabla_\theta\mathcal{L}_{\text{Q}}(\theta) = \mathbb{E}_{s,\epsilon}\left[\nabla_a Q_\phi(s,a) \cdot \nabla_\theta a\right] \tag{42}$$

$$= \mathbb{E}_{s,\epsilon}\left[\nabla_a Q_\phi(s,a) \cdot (-\nabla_\theta u_\theta(\epsilon, 0, 1; s))\right]. \tag{43}$$

Unlike diffusion-based policy parameterizations that require backpropagating through many iterative denoising steps (BPTT), the one-step mapping $a = \epsilon - u_\theta(\epsilon, 0, 1; s)$ is a single differentiable transformation. Thus, $\nabla_\theta a$ is computed in one step without temporal unrolling, making the actor update significantly faster, more training-friendly.

**Gradient of the FBC term.** The FBC objective is

$$\mathcal{L}_{\text{FBC}}(\theta) = \mathbb{E}_{s,a,t,r,\epsilon}\left[\|u_\theta(a_t, r, t; s) - \text{sg}(u_{\text{tgt}})\|_2^2\right], \tag{44}$$

where $u_{tgt}$ is stop-gradient $\text{sg}(\cdot)$:

$$u_{\text{tgt}} = v_t - (t - r)\left(v_t \cdot \partial_{a_t} u_\theta + \partial_t u_\theta\right), \qquad a_t = (1 - t)a + t\epsilon, \qquad v_t = \epsilon - a \tag{45}$$

Because of sg(.), the target is treated as constant when differentiating. Thus

$$\nabla_\theta\mathcal{L}_{\text{FBC}}(\theta) = 2\,\mathbb{E}_{s,a,t,r,\epsilon}\left[(u_\theta(a_t, r, t; s) - u_{\text{tgt}}) \cdot \nabla_\theta u_\theta(a_t, r, t; s)\right]. \tag{46}$$

**Full OFQL actor gradient.** Combining Eqs. 42–46:

$$\nabla_\theta \mathcal{L}(\theta) = \nabla_\theta \mathcal{L}_{\text{FBC}}(\theta) - \alpha \nabla_\theta \mathcal{L}_{\text{Q}}(\theta) \tag{47}$$

$$= 2\, \mathbb{E}_{s,a,t,r,\epsilon}[(u_\theta(a_t, r, t; s) - u_{\text{tgt}}) \cdot \nabla_\theta u_\theta(a_t, r, t; s)] \tag{48}$$

$$+ \alpha\, \mathbb{E}_{s,\epsilon}[\nabla_a Q_\phi(s, a) \cdot \nabla_\theta u_\theta(\epsilon, 0, 1; s)]. \tag{49}$$

**Interpretation.** The first term regularizes the policy toward the behavior distribution by matching average velocities, while the second term regularizes the policy to maximize the critic value through the differentiable one-step action mapping. Thus, OFQL simultaneously achieves behavior regularization and return maximization.

## J   EVALUATING OFQL IN HIGH-DIMENSIONAL ACTION ROBOTIC MANIPULATION

To further evaluate OFQL in high-dimensional action spaces, we conducted additional experiments on the D4RL Adroit benchmark, which features 24-dimensional control using a dexterous robotic hand. We evaluated two standard tasks—adroit-pen-human and adroit-pen-cloned—where the objective is to manipulate a pen to match a target orientation using a 24-DoF hand. This domain is particularly challenging due to noisy human demonstrations, sparse rewards, and the high-dimensional action manifold. Normalized returns, following the evaluation protocol of Fu et al. (2020), are reported below:

| Task | BC | DQL | OFQL (ours) |
|------|-----|------|-------------|
| adroit-pen-human | 71.0 | 75.7±9.0 | **79.5±9.5** |
| adroit-pen-cloned | 52.0 | 60.8±11.8 | **62.3±10.3** |

The results show that OFQL consistently outperforms both BC and DQL in high-dimensional manipulation tasks, demonstrating strong robustness and effectiveness in complex dexterous control settings.

## K   FEASIBILITY ON VISUAL OBSERVATION SETTING

To demonstrate the feasibility of OFQL in the visual-observation setting, we evaluate it on two OGBench (Park et al., 2024) visual manipulation tasks that require reasoning over high-dimensional image observations (64×64×3): `visual-scene-singletask-task1-v0` (moderate difficulty) and `visual-cube-double-play-singletask-task1-v0` (hard). We adopt the small IMPALA encoder (following FQL (Park et al., 2025)) for embedding the image observation to the latent state and use simple concatenation for state conditioning. Task success rates are reported below:

| Task | FBRAC | FQL | OFQL (ours) |
|------|-------|-----|-------------|
| visual-scene-singletask-task1-v0 | 46.0±4.0 | **98.0±3.0** | 54.0±9.0 |
| visual-cube-double-play-singletask-task1-v0 | 6.0±2.0 | **21.0±11.0** | 8.0±3.0 |

These results show that OFQL remains functional in visual settings, but its performance lags behind stronger visual baselines, indicating that additional architectural and algorithmic considerations are necessary for competitive results in high-dimensional pixel-based domains.

There are several key challenges when extending OFQL to high-dimensional input scenarios such as image-based observations.

First, the learning objectives become tightly coupled. Unlike low-dimensional state spaces, visual tasks require the policy to jointly learn (i) accurate Q-values, (ii) flow-based behavior regularization, and (iii) a stable and expressive visual representation. These components are deeply interdependent: noise or instability in the visual encoder propagates into Q-value estimation and flow predictions,

while inaccuracies in the critic or policy can, in turn, misguide the encoder. This tight coupling makes the overall optimization process considerably more fragile compared to low-dimensional settings.

Second, conditioning high-dimensional latent features into the flow network is non-trivial. Simple concatenation of visual latents with the noise vector may be insufficient. High-dimensional representations often require more structured fusion strategies—e.g., FiLM layers, cross-attention,..—to ensure the visual features meaningfully influence the learned flow direction. Without proper conditioning, the policy may ignore or underutilize visual information.

Third, representation quality may becomes a bottleneck. Lightweight or general-purpose encoders may fail to capture task-relevant spatial and semantic cues required for precise action prediction. Stronger or task-specific visual backbones, domain augmentations, or auxiliary representation-learning losses (e.g., predictive coding (Nguyen et al., 2021b), contrastive learning (Luu et al., 2022; Chen et al., 2020; Nguyen et al., 2023),...) may be necessary to maintain stable training.

Overall, extending OFQL to visual domains will likely require more robust encoders, improved conditioning strategies, and additional guidance signals to ensure that visual features effectively support flow-based policy learning which is an interesting direction for future work.

## L  HANDLING OUT-OF-DISTRIBUTION STATES: OFQL VS. DQL

Across all benchmarks, OFQL consistently attains higher average returns than DQL. This suggests improved robustness to out-of-distribution (OOD) states, since the average return is determined by the policy's interaction with the real environment, where trajectories often drift outside the trained distribution. A policy that performs better in these interactions is implicitly better at handling such OOD states.

To further support this, we compute the mean squared error between the actions produced by the trained policy (trained on medium / medium-replay datasets) and the expert actions on the expert dataset. Let's denotes this MSE as OOD-MSE. This metric measures how well the learned policy aligns with the expert policy under the expert state distribution, which is largely out-of-distribution relative to the training data. A lower OOD-MSE therefore indicates stronger generalization to unseen or OOD states. we provide OOD-MSE on the HalfCheetah dataset as below.

| Metric (Dataset) | OOD-MSE (Medium) | OOD-MSE (Medium-Replay) |
|---|---|---|
| OFQL | **0.458** | **0.560** |
| DQL | 0.462 | 0.582 |

The results presented in the table above show that OFQL consistently achieves lower OOD-MSE than its DQL counterpart, demonstrating that OFQL generalizes more effectively to unseen or out-of-distribution states.

## M  ABLATION ON TIME-SAMPLING DISTRIBUTION

We evaluate the effect of the time-sampling distribution in OFQL by comparing uniform sampling against a logit-normal distribution. An ablation on HalfCheetah is summarized below:

| Time-Sampling | Uniform | Logit-Normal |
|---|---|---|
| Medium-Expert | 94.5±0.5 | **95.2±0.4** |
| Medium | 61.1±0.1 | **63.8±0.1** |
| Medium-Replay | **51.7±0.2** | 51.2±0.1 |

Overall performance is similar across the two strategies, though the logit-normal distribution yields a slight improvement on some datasets. These results show that OFQL remains robust under different time-sampling strategies, and performance is not highly sensitive to the precise tuning of this hyperparameter. In practice, we use logit-normal parameters ($\mu = -0.4, \sigma = 1.0$) as the default.

## N  BASELINES REPRODUCIBILITY

**Baseline Result.** For DQL and FQL on AntMaze, we directly report the results from the original papers. For other baselines—including BC, TD3-BC, IQL, Diffuser, DD, EDP, and IDQL—we use results from the broadly accepted and standardized reimplementation CleanDiffuser (Dong et al., 2024). For details on training and evaluation procedures, we refer readers to the corresponding papers.

For the FQL on Locomotion and Kitchen, the official FQL implementation does not support the D4RL Locomotion or Kitchen domains. To ensure fair comparison, we extend the official JAX codebase to support these environments and additionally implement a PyTorch version of FQL within the OFQL framework (our implementation is based on PyTorch). We follow the recommendations from the official FQL repository and paper: we use the normalized-Q setting and perform a hyperparameter search over $\alpha \in \{0.03, 0.1, 0.3, 1, 3, 10\}$, as described in Appendix C of Park et al. (2025). For network architecture, we search over MLP sizes $[512, 512, 512, 512]$ and $[256, 256, 256, 256]$. We run both our extended JAX version and our PyTorch implementation and report the best-performing results. For speed measurements, we use the PyTorch version to avoid framework-level differences (JAX vs. PyTorch).

**Diffusion Steps.**  We follow the standard diffusion-step settings recommended by each baseline: DQL (5 steps), IDQL (5 steps), EDP (15 steps), and the Flow Model used in FQL (10 steps). These configurations align with the settings reported in the respective papers or official repositories.

## O  ON THE MEANFLOW IDENTITY

For completeness, the derivation from (Geng et al., 2025) is revisited to provide a clear understanding of how MeanFlow Identity can be used to calculate the target average velocity.

Let's consider the no-condition generation case (no state condition) for simplicity. The *average velocity* is defined as the displacement between two timesteps $t$ and $r$ divided by the time interval:

$$u(a_t, r, t) \triangleq \frac{1}{t-r} \int_r^t v(a_\tau, \tau) d\tau. \tag{50}$$

Here, $u$ denotes the average velocity, $v$ the instantaneous velocity (i.e., marginal velocity), and $a$ is the noise action. As $r \to t$, $u$ converges to $v$.

Our purpose is to approximate $u$ using a neural network, enabling single-step generation (i.e., $r = 0, t = 1$), unlike methods based on marginal velocity (a.k.a instantaneous velocity), which require time integration at inference. Direct training with $u$ is impractical due to the integral; instead, the definition of $u$ is manipulated to derive a tractable optimization target.

**The MeanFlow Identity.** To facilitate training, the equation for $u$ is rewritten as:

$$(t-r)u(a_t, r, t) = \int_r^t v(a_\tau, \tau) d\tau. \tag{51}$$

Differentiating both sides with respect to $t$ gives:

$$\frac{d}{dt}(t-r)u(a_t, r, t) = \frac{d}{dt} \int_r^t v(a_\tau, \tau) d\tau \implies u(a_t, r, t) + (t-r)\frac{d}{dt}u(a_t, r, t) = v(a_t, t) \tag{52}$$

Rearranging, the *MeanFlow Identity* is achieved:

$$u(a_t, r, t) = v(a_t, t) - (t-r)\frac{d}{dt}u(a_t, r, t) \tag{53}$$

This identity links $u$ and $v$, providing a target for training a neural network. The next step is to compute the time derivative of $u$.

**Computing the Time Derivative.** To compute $\frac{d}{dt}u$, we expand the total derivative:

$$\frac{d}{dt}u(a_t, r, t) = \frac{da_t}{dt}\partial_{a_t}u + \frac{dr}{dt}\partial_r u + \frac{dt}{dt}\partial_t u \tag{54}$$

Using $\frac{da_t}{dt} = v(a_t, t)$, $\frac{dr}{dt} = 0$, and $\frac{dt}{dt} = 1$, we obtain:

$$\frac{d}{dt} u(a_t, r, t) = v(a_t, t)\partial_z u + \partial_t u$$

This shows that the total derivative of $u$ is computed as the Jacobian-vector product (JVP) of the network's Jacobian and the tangent vector $[v, 0, 1]$.

Notably, the MeanFlow Identity (Eq. 53) is mathematical equivalent to Eq. 51 (Geng et al., 2025).

We train the policy network by conditioning on the state $s$, parameterizing $u_\theta(a_t, r, t; s)$, and applying the *MeanFlow Identity* to define the optimization target.

## P  LLM USAGE

In preparing this paper, we used Large Language Models (LLMs) solely as an assistive tool for grammar checking and polishing text. The LLMs were not involved in research ideation, experimental design, data analysis, or substantive content generation. All research ideas, methods, analyses, and conclusions are the authors' own.

