# OpenReview forum: "One-Step Flow Q-Learning: Addressing the Diffusion Policy Bottleneck in Offline Reinforcement Learning"
_ICLR.cc/2026/Conference — ICLR 2026 Poster_

### Official Review · Reviewer_43m1 · 2025-10-30

**Soundness:** 3
**Presentation:** 4
**Contribution:** 3
**Rating:** 6
**Confidence:** 3

**Summary:**

To address the slow multi-step denoising and unstable optimization inherent in diffusion-based policies, this paper introduces One-Step Flow Q-Learning (OFQL). OFQL reformulates the diffusion denoising process within the Flow Matching (FM) framework and learns an average velocity field that enables direct one-step action generation. Experiments across diverse D4RL tasks demonstrate that OFQL achieves the highest average normalized scores among all compared methods. Moreover, its one-step sampling design substantially improves both training and inference efficiency.

**Strengths:**

OFQL is an efficient reinforcement learning algorithm that introduces average velocity fields within the Flow Matching framework, enabling it to model complex policy distributions without relying on distillation procedures or auxiliary networks. The method offers a unified training–inference pipeline, using the same one-step model consistently in both phases. Empirically, OFQL outperforms DQL and other strong baselines in terms of both policy performance and computational efficiency.

**Weaknesses:**

1.	OFQL relies on the MeanFlow Identity to enable one-step sampling for the learned policy. However, the Jacobian–vector product computation in Eq. (11) may become computationally demanding for large-scale models.

2.	As acknowledged in the paper’s limitations, it remains unclear whether OFQL can scale to high-dimensional action spaces (e.g., humanoid control or vision-based RL). Moreover, the stability of the proposed one-step policy under non-stationary or online settings has not been investigated.

3.	The paper provides no formal analysis establishing the expressive equivalence between the average-velocity one-step formulation and traditional multi-step diffusion policies.

**Questions:**

1.	Does learning an average velocity field constrain the representational power compared to DDPM’s full reverse process?

2.	How sensitive is OFQL to flow ratio and time-sampling distribution?

3.	Does one-step flow matching better handle out-of-distribution states than diffusion-based DQL?

4.	How exactly does the Q-gradient interact with flow learning?

---

> ### Author Response · Authors · 2025-11-26
> **Rebuttal by Authors Part #1**
>
> Thank you for the thorough review and constructive feedback on our work. Below, we address your questions in detail and provide the corresponding explanations or actions taken to mitigate the identified weaknesses.
>
> **Question 1. Does learning an average velocity field constrain the representational power compared to DDPM’s full reverse process?**
>
> Learning an average-velocity field does not restrict representational capacity. It is simply an alternative parameterization of the marginal velocity field used in flow matching. As a result, its expressive power is comparable to standard flow-matching models, which are well known to match or closely approximate the generative expressiveness of DDPMs.
>
> To further verify this, we include an ablation comparing the expressive ability of the average-velocity model and a DDPM baseline on controlled toy examples (refer to Appendix C). The results show that the average-velocity model achieves comparable generation quality—even with a single sampling step—supporting that  average-velocity field does not restrict representational capacity.
>
> For completeness, we note that the key benefit of the average-velocity formulation is not improved expressivity but its one-step sampling capability. This eliminates backpropagation through long diffusion trajectories in the actor–critic loop, yielding substantially faster and more robust policy training.
>
> > Changes Added in the Revision:
> > + Add an experiment comparing the expressive ability of the average-velocity model and a DDPM baseline on controlled toy examples (Appendix C)
>
>
>
>
> **Question 2. How sensitive is OFQL to flow ratio and time-sampling distribution?**
>
>
> OFQL exhibits low sensitivity to both the flow ratio and the time-sampling distribution.
>
> **Flow ratio.** In practice, setting the flow ratio to 0.5 works well across most D4RL tasks. Our ablation study (Ablation on flow ratio) shows that values in the range [0.25, 0.75] yield stable and consistent policy learning, indicating low sensitivity to this hyperparameter.
>
> **Time-sampling distribution.** OFQL performs well under both uniform and logit-normal time-sampling. While the overall trends are similar, the logit-normal distribution provides a slight improvement on some datasets. We can use parameters (–0.4, 1.0) as the default. The ablation on the HalfCheetah comparing the two distributions is summarized below:
>
>
> | Time-sampling | Uniform        | Logit-normal   |
> | ------------- | -------------- | -------------- |
> | Medium-Expert | 94.5 ± 0.5     | **95.2 ± 0.4** |
> | Medium        | 61.1 ± 0.1     | **63.8 ± 0.1** |
> | Medium-Replay | **51.7 ± 0.2** | 51.2 ± 0.1     |
>
>
> These results demonstrate that OFQL remains robust across a wide range of choices, and performance is not heavily dependent on precise tuning of either hyperparameter.
>
> > Changes Added in the Revision:
> >+ Add appendix M.ABLATION ON TIME-SAMPLING DISTRIBUTION
>
>
> **Question 3. Does one-step flow matching better handle out-of-distribution states than diffusion-based DQL?**
>
> Yes—empirically. Across all benchmarks, OFQL consistently attains higher average returns than DQL. This suggests improved robustness to out-of-distribution (OOD) states, since the average return is determined by the policy’s interaction with the real environment, where trajectories often drift outside the trained distribution. A policy that performs better in these interactions is implicitly better at handling such OOD states.
>
> To further support this, we compute the mean squared error between the actions produced by the trained policy (trained on medium / medium-replay datasets) and the expert actions on the expert dataset. Let's denote this MSE as OOD-MSE. This metric measures how well the learned policy aligns with the expert policy under the expert state distribution, which is largely out-of-distribution relative to the training data. A lower OOD-MSE therefore indicates stronger generalization to unseen or OOD states. We provide OOD-MSE on the HalfCheetah dataset as below.
>
> | Metric(Dataset)|           OOD_MSE    (Medium)      |      OOD_MSE(Medium-Replay)|
> |--|-----------------------|----------------------------|
> OFQL|    **0.458**           | **0.560**
> DQL  |        0.462      |             0.582
>
> The results presented in the table above show that OFQL consistently achieves lower OOD-MSE than its DQL counterpart, demonstrating that OFQL generalizes more effectively to unseen or out-of-distribution states.
>
> > **Changes Added in the Revision:**
> >+ Add appendix L. Handling Out-of-Distribution States: OFQL vs. DQL

---

> ### Author Response · Authors · 2025-11-26
> **Rebuttal by Authors Part 2**
>
> **4. How exactly does the Q-gradient interact with flow learning?**
>
> We add the gradient analysis of OFQL actor loss to the paper, which explains the exact gradients and their interactions. A brief summary is as below:
>
>
> $L(θ)=L_{FBC}(θ)−αL_{Q}(θ)$
>
> $\nabla_θ L(θ)=2E_{s,a,t,r,ϵ}[(u_θ(a_t,r,t;s)−u_{tgt})⋅\nabla_θu_θ(a_t,r,t;s)]+αE_{s,ϵ}[∇_aQ_ϕ(s,a)⋅∇_θu_θ(ϵ,0,1;s)].$
>
>
>
> The first term regularizes the policy toward the behavior policy by aligning average velocities. The second term—the Q-gradient—steers the policy toward actions that maximize the critic value through the differentiable one-step action mapping (which is the approximate flow matching endpoint map). Consequently, OFQL jointly achieves behavior regularization and return maximization without requiring backpropagation through time, while still supporting complex action distributions by following the nonlinear endpoint map induced by flow-matching dynamics.
>
> For a more detailed explanation and derivation, we refer to related changes.
> > Changes Added in the Revision:
> >+ Add appendix I.GRADIENT ANALYSIS OF OFQL ACTOR LOSS
> >+ Add Appendix H. FORMAL JUSTIFICATION OF AVERAGE-VELOCITY LEARNING ENCOURAGES THE LEARNED ONE-STEP POLICY TO STAY CLOSE TO THE BEHAVIOR POLICY
>
>
> **Addressing Weakness 1: the Jacobian–vector product computation on large-scale models**
> In OFQL, the JVP is lightweight: computing $du$ requires only one extra backward pass, and because it is stop-gradient w.r.t. $\theta$, it introduces no higher-order gradients and adds no cost to parameter backpropagation. The JVP backward pass is similar to a standard backward pass and can even be cheaper since it backpropagates only to inputs. For an MLP-based policy such as the one used in OFQL, this overhead is negligible. On a much larger 131M-parameter model, a $\sim 16$% increase in wall-clock time is observed, offering a practical reference for the expected cost.
>
> **Addressing Weakness 2: scale to high-dimensional action spaces or vision-based RL**
>
> This concern was also raised by Reviewers opYF and QtuL. We restate our response here for your convenience.
>
> **High-dimensional action.** To further evaluate OFQL in high-dimensional action spaces, we conducted additional experiments on the D4RL Adroit benchmark, which features 24-dimensional control using a dexterous robotic hand.We evaluated two standard tasks—adroit-pen-human and adroit-pen-cloned—where the objective is to manipulate a pen to match a target orientation using a 24-DoF hand. Normalized returns are reported below:
> ||BC|DQL|OFQL|
> |--|--|--|--|
> |adroit-pen-human|71.0|75.7+-9.0|**79.5+-9.5**
> |adroit-pen-cloned|52.0|60.8+-11.8|**62.3+-10.3**
>
> The results show that OFQL consistently outperforms both BC and DQL in
> high-dimensional manipulation tasks, demonstrating strong robustness and
> effectiveness in complex dexterous control settings.
>
> **High-dimensional visual observation.**  We additionally evaluate OFQL in high-dimensional input settings using the OGBench benchmark (2 visual tasks). The results show that OFQL remains functional with raw visual observations; however, its performance is still limited. We argue that achieving strong results in visual offline RL likely requires additional architectural and training design, such as stronger visual encoders, improved conditioning strategies, or representation regularization.
> |  |FBRAC|FQL|OFQL|
> |--|--|--|--|
> |visual-scene-play-singletask-task1-v0|46.0+-4.0|**98.0+-3.0**|54.0+-9.0|
> |visual-cube-double-play-singletask-task1-v0|6.0+-2.0|**21.0+-11.0**|8.0+-3.0|
>
> We hope these results further strengthen the paper and provide readers with a more comprehensive understanding of OFQL’s applicability across both high-dimensional action and visual input domains.
>
> For further details, we refer you to the revisions outlined below.
>
> > Changes Added in the Revision:
> > + Add Appendix J. Evaluating OFQL in High-Dimensional Action Robotic Manipulation
> > + Add Appendix K. Feasibility on visual observation

---

> ### Author Response · Authors · 2025-11-27
> **Rebuttal by Authors Part 3**
>
> **Addressing Weakness 3: formal analysis on the expressivity of OFQL**
>
> This concern aligns with Reviewer 8FUi’s Question 1, and we refer you to our detailed response there. Briefly, learning the average velocity preserves the expressive capacity of flow matching, which is well established in generative modeling to have expressivity comparable to DDPM. Consequently, OFQL is expected to maintain similar expressive power to DDPM-based policies. Our response to your Question 1 further substantiates this claim.
>
> Below are the related additions in the revision that provide the requested formal analysis.
>
> > Related Changes Added in the Revision:
> > + Add Appendix G. FORMAL JUSTIFICATION OF ACTION ACCURACY PRESERVATION IN ONE-STEP GENERATION THROUGH AVERAGE VELOCITY FIELD
> >+ Add Appendix H. FORMAL JUSTIFICATION OF AVERAGE-VELOCITY LEARNING ENCOURAGES THE LEARNED ONE-STEP POLICY TO STAY CLOSE TO THE BEHAVIOR POLICY
> >+ Add an experiment comparing the expressive ability of the average-velocity model and a DDPM baseline on controlled toy examples (Appendix C)
> -----
> We once again sincerely thank the reviewer for the constructive comments and insightful questions, which have helped us significantly improve the quality of the paper. We hope that our clarifications adequately address your concerns and the identified weaknesses, and we would be grateful if you would consider this in your updated rating.

---

### Official Review · Reviewer_QtuL · 2025-11-01

**Soundness:** 2
**Presentation:** 3
**Contribution:** 2
**Rating:** 4
**Confidence:** 4

**Summary:**

This paper proposes One-Step Flow-Q-Learning, an offline RL algorithm that enables one-step action generation during both training and inference. The method is closely related to DQL but leverages the average velocity parametrization following MeanFlow. The action sampling in this parametrization can be done in a single step, which reduces inference cost and avoids the huge computation and memory costs of backpropagation through the multi-step sampling chain. Experiments on the D4RL benchmark show that OFQL surpasses all other included baselines in overall performance.

**Strengths:**

1. The method shows empirical advantages in policy performance, training speed and inference time.
2. The paper is easy to follow.

**Weaknesses:**

1. The proposed method lacks novelty. The only main difference between the proposed method and DQL is replacing the diffusion loss in actor training with a MeanFlow loss.
2. The experiments are not adequate. Only results on state-based D4RL tasks are included, and no visual observation task results are reported.
3. The argument in Lines 262-264 is not clear. Flow matching cannot "in principle, enable one-step generation", as the sampling trajectory is straight only when the target distribution is a delta distribution or when rectification or similar techniques have been used. The following sentences in this paragraph are accurate.

**Questions:**

1. How many diffusion steps are used for the multi-step diffusion policy baselines? Is the number aligned with the original papers?
2. Can the proposed method be extended to visual observation tasks? Are there any challenges for the method in high-dimensional input scenarios?

---

> ### Author Response · Authors · 2025-11-26
> **Rebuttal by Authors Part #1**
>
> Thank you for the thorough review and constructive feedback on our work. Below, we address your questions in detail and provide the corresponding explanations or actions taken to mitigate the identified weaknesses.
>
> **Question 1. How many diffusion steps are used for the multi-step diffusion policy baselines? Is the number aligned with the original papers?**
>
> We report results following the standard diffusion-step settings recommended by each baseline: DQL (5 steps), IDQL (5 steps), EDP(15 steps), and  FQL ( the Flow Model uses 10 steps).  These
> configurations align with the settings reported in the respective papers.
>
> > Changes Added in the Revision:
> > + Add Appendix. N BASELINES REPRODUCIBILITY to detail diffusion steps and how to get baseline results.
>
>
> **Question 2.1. Can the proposed method be extended to visual observation tasks?**
>
> Yes, the proposed method can be extended to visual-observation tasks, although this setting requires additional care/investigation, as we acknowledged in the Appendix (Limitations). The simplest extension is to encode each image into a latent representation and condition the average-velocity model on this latent. To demonstrate feasibility, we evaluate OFQL on two OGBench (#1) visual manipulation tasks that challenge the agent with high-dimensional observations (64×64×3): the visual-scene-singletask-task1-v0 (moderate difficulty) and visual-cube-double-play-singletask-task1-v0 (hard). We adopt the small IMPALA encoder, following FQL, for image encoding and use simple concatenation for conditioning. The success rate on these tasks is reported below:
>
>
> |  |FBRAC|FQL|OFQL|
> |--|--|--|--|
> |visual-scene-play-singletask-task1-v0|46.0+-4.0|**98.0+-3.0**|54.0+-9.0|
> |visual-cube-double-play-singletask-task1-v0|6.0+-2.0|**21.0+-11.0**|8.0+-3.0|
>
> These results show that OFQL remains functional in visual settings, but performance is still low, indicating that additional design considerations are necessary for strong results.
>
> **Question 2.2. Are there any challenges for the method in high-dimensional input scenarios?**
>
> There are several key challenges when extending OFQL to high-dimensional input scenarios such as image-based observations.
>
> First, the learning objectives become tightly coupled. Unlike low-dimensional state spaces, visual tasks require the policy to jointly learn (i) accurate Q-values, (ii) flow-based behavior regularization, and (iii) a stable and expressive visual representation. These components are deeply interdependent: noise or instability in the visual encoder propagates into Q-value estimation and flow predictions, while inaccuracies in the critic or policy can, in turn, misguide the encoder. This tight coupling makes the overall optimization process considerably more fragile compared to low-dimensional settings.
>
> Second, conditioning high-dimensional latent features into the flow network is non-trivial. Simple concatenation of visual latents with the noise vector may be insufficient. High-dimensional representations often require more structured fusion strategies—e.g., FiLM layers, cross-attention,..—to ensure the visual features meaningfully influence the learned flow direction. Without proper conditioning, the policy may ignore or underutilize visual information.
>
> Third, representation quality may become a bottleneck. Lightweight or general-purpose encoders may fail to capture task-relevant spatial and semantic cues required for precise action prediction. Stronger or task-specific visual backbones, domain augmentations, or auxiliary representation-learning losses may be necessary to maintain stable training.
>
> Overall, extending OFQL to visual domains will likely require more robust encoders, improved conditioning strategies, and additional guidance signals to ensure that visual features effectively support flow-based policy learning, which is an interesting direction for future work.
>
> > Changes Added in the Revision:
> > + Add Appendix K. Feasibility on visual observation
>
> #1. OGBench https://seohong.me/projects/ogbench/

---

> ### Author Response · Authors · 2025-11-26
> **Rebuttal by Authors Part 2**
>
> **Address Weakness 1. The simple replacement of the diffusion loss**
>
> Our modification is minimal, but this is intentional and aligns with our motivation: to identify a simple and direct change that enables reliable one-step action generation in reinforcement learning. Achieving **a simple yet effective one-step framework is non-trivial** due to the inherently unstable nature of RL optimization, though. Prior work has shown that naïvely applying one-step generative models often fails in the one-step regime (e.g., Consistency-AC (Table 3)(#2) ), which is why existing solutions typically rely on multi-stage distillation or multi-phase training, resulting in more complex algorithms.
>
> Additionally, OFQL operates in the RL setting and not just pure supervise learning (e.g, MeanFlow), where optimization involves interactions between the policy, Q-function, and the average-velocity objective. The use of the Mean Flow identity allows us to compute a tractable average-velocity field, but the policy is updated not only through this objective but also through the Q-function—an optimization structure fundamentally different from supervised training. We have clarified these distinctions in the Related Work
>
>
> To further address the reviewer’s concern regarding novelty and provide further insight in the RL setting, we have expanded the paper with:
> (i) formal justifications showing how average-velocity learning connects policy learning to Flow Matching dynamics (see Reviewer8FUi/Question 1), and
> (ii) a gradient-flow analysis illustrating how the policy and Q-function interact during training (see Reviewer 43m1/Question 4)
>
> We hope these additions strengthen the conceptual understanding of why the proposed one-step formulation works within RL
>
> > Changes Added in the Revision:
> > + Add Appendix G. FORMAL JUSTIFICATION OF ACTION ACCURACY PRESERVATION IN ONE-STEP GENERATION THROUGH AVERAGE VELOCITY FIELD
> > + Add Appendix H. FORMAL JUSTIFICATION OF AVERAGE-VELOCITY LEARNING ENCOURAGES THE LEARNED ONE-STEP POLICY TO STAY CLOSE TO THE BEHAVIOR POLICY
> > + Add Appendix I. GRADIENT ANALYSIS OF OFQL ACTOR LOSS
>
> **Address Weakness 2. State-Based D4RL Usage and Visual Observation Experiments**
>
> We intentionally focus on state-based D4RL tasks because they remove confounding factors ( visual feature extraction, visual fusion, architecture-specific biases...). This allows us to isolate and analyze the algorithmic contribution of OFQL more accurately. Moreover, D4RL is not a single dataset but a comprehensive benchmark suite. In our experiments, we evaluate on 15 diverse datasets/tasks (covering narrow and biased data distributions, suboptimal demonstrations, multi-task and undirected data, sparse-reward regimes...) These tasks are widely used by prior offline RL methods and are appropriate for assessing core algorithmic behavior.
>
> To further strengthen the evaluation, we have added experiments on visual-observation tasks (your question 2) and high-dimensional action-space environments (See reviewer opYF, Weakness). These additional results complement the state-based analysis to address the reviewer’s concern
>
> > Changes Added in the Revision:
> > + Add Appendix J. Evaluating OFQL in High-Dimensional Action Robotic Manipulation
> > + Add Appendix K. Feasibility on visual observation
>
> **Addressing Weakness 3. The argument in Lines 262-264 is not clear.**
>
> We updated the sentence to make it clearer in the main paper, following your suggestion
>
> > Changes Added in the Revision:
> > + Update main paper (lines 261-265)
>
>
> \#2. Consistency-AC https://arxiv.org/pdf/2309.16984
>
> ---
>
> **We once again sincerely thank the reviewer for the careful reading and insightful feedback. We hope that our clarifications adequately address your concerns, and we would be grateful if you might consider a higher rating**

---

### Official Review · Reviewer_8FUi · 2025-11-03

**Soundness:** 3
**Presentation:** 3
**Contribution:** 3
**Rating:** 8
**Confidence:** 4

**Summary:**

This paper proposes **One-Step Flow Q-Learning (OFQL)**, a novel framework for offline reinforcement learning that reformulates Diffusion Q-Learning (DQL) within the **Flow Matching (FM)** paradigm. By learning an **average velocity field** instead of a marginal one, OFQL enables accurate **one-step action generation** during both training and inference—eliminating the need for multi-step denoising and recursive backpropagation. This design substantially improves training and inference efficiency while maintaining, and even improving, performance. The authors demonstrate strong results across D4RL benchmarks.

**Strengths:**

* **Clear conceptual advancement:** Reformulating DQL under the flow-matching framework and introducing an average velocity field is a novel and elegant idea that directly addresses the core inefficiency of multi-step denoising.
* **Simplicity and effectiveness:** Unlike prior one-step approaches that depend on auxiliary modules or policy distillation, OFQL remains conceptually clean while achieving superior results.
* **Strong empirical results:** The method outperforms DQL and other diffusion-based baselines by a significant margin on D4RL, demonstrating both **efficiency** and **robustness**.
* **Illustrative toy example:** The toy experiment effectively clarifies the intuition behind the average velocity field and supports the main claim.
* **Readable and well-organized:** The paper is well-written, clearly structured, and easy to follow even for readers not deeply familiar with flow-matching methods.

**Weaknesses:**

* The theoretical justification for why learning an **average velocity field** leads to better one-step performance could be elaborated further. Currently, the paper provides an intuitive explanation but lacks a deeper analytical connection to diffusion dynamics.

**Questions:**

1. Could the authors provide a more formal justification for why **average velocity learning** preserves accuracy in one-step action generation?
2. Are there scenarios (e.g., highly multimodal action distributions) where the **average velocity** assumption might underperform?

Typo: citation in line 151

---

> ### Author Response · Authors · 2025-11-26
> **Rebuttal by Authors**
>
> Thank you for the detailed review and constructive feedback on our work. Below, we provide formal justification that directly addresses Question 1 and the associated weakness. We also provide our response to Question 2, clarifying the potential limitations of the average-velocity assumption.
>
> **Question 1, Weakness. Formal justification for why average-velocity learning preserves one-step accuracy and connection to diffusion dynamics**
>
> We have expanded the main paper (Lines 277--298) and the Appendix. G and  Appendix. H to give more explanation and include formal justifications. The core (summary) argument is as follows.
>
> **Perfect learning.** Under the Flow Matching (FM) dynamics, the target behavior distribution $\mu(\cdot \mid s)$ is the push-forward of a base Gaussian under the ground-truth endpoint map $T^\star$:
>
> $\mu(\cdot \mid s) = (T^\star)_\\# \mathcal{N}(0, I).$
>
> By definition, the average velocity is a functional of the underlying instantaneous velocity field $v^\star$. Therefore, if the average velocity is learned exactly, the learned endpoint map $T_\theta$ produced by our one-step generator (Eq.~12) coincides with $T^\star$. Consequently, the learned policy distribution satisfies
>
> $\pi_\theta (\cdot \mid s) = (T_\theta)_\\# \mathcal{N}(0, I) $
>
> $= (T^\star)_\\# \mathcal{N}(0, I) = \mu (\cdot \mid s)$
>
>
>
> which guarantees accurate one-step action generation.
>
> **Imperfect learning.** We show that minimizing the average-velocity loss $L_{FBC^\\star} (\theta)$ drives the resulting one-step policy $\pi_\theta(\cdot\mid s)$ toward the behavior policy $\mu(\cdot\mid s)$, thereby performing behavior cloning.  Importantly, this behavior cloning still preserves the ability to model complex, multimodal action distributions through the nonlinear transport map inherited from Flow Matching.
>
> Summarily, we show that $\pi_\theta$ and $\mu$ are fully determined by their $u_\theta, u^*$ , respectively and the theoretical average-velocity loss satisfies
>
> $L(u_\theta, u^\star) \geq p_{01} E_{s} [W_2 (\pi_\theta(\cdot\mid s),\mu(\cdot\mid s))^2] $
>
> Thus, as $\mathcal{L}(u_\theta, u^\star) \to 0$, we obtain $W_2(\pi_\theta, \mu) \to 0$ in expectation over $s$. In effect, average-velocity learning regularizes $\pi_\theta(\cdot\mid s)$ toward the behavior distribution $\mu(\cdot\mid s)$, while preserving the ability to represent complex, multimodal actions through the nonlinear endpoint map induced by FM dynamics.
>
> The above provides a brief summary for your convenience; for full derivations and detailed explanations, please refer to the revised manuscript.
> > Changes Added in the Revision:
> >  + Add Appendix G. FORMAL JUSTIFICATION OF ACTION ACCURACY PRESERVATION IN ONE-STEP GENERATION THROUGH AVERAGE VELOCITY FIELD
> >  + Add Appendix H. FORMAL JUSTIFICATION OF AVERAGE-VELOCITY LEARNING ENCOURAGES THE LEARNED ONE-STEP POLICY TO STAY CLOSE TO THE BEHAVIOR POLICY
> > + Add a summary, explaining the formal justifications about average velocity learning, in the main paper (Design One-Step Policy) and link these to the formal justifications in Appendices G, H, and O.
>
> **Question 2. Failure case of the average-velocity assumption.**
>
> In theory, the average velocity is merely a functional of the underlying instantaneous velocity field. Thus, if the instantaneous velocity can be learned accurately, the resulting average velocity will also be accurate. In this sense, its success entirely depends on the quality of the implicit learned instantaneous velocity governs the FM dynamics and is expected to work well in high-dimensional settings, where Flow Matching is well known to perform strongly.
>
> In practice, a key failure mode may arise. The training loss estimates the instantaneous velocity on-the-fly from conditional velocities. When this estimation is inaccurate, the resulting instantaneous velocity contains systematic errors that propagate into the learned average velocity. This compounding effect can hinder effective learning and explains the degraded performance observed on some datasets when not caring about this aspect (See ablation on flow ratio).
>
> To address this, OFQL biases training toward learning a more accurate instantaneous velocity. Concretely, we use a controlled flow ratio that prioritizes instantaneous-velocity learning while still bootstrapping into the average velocity. In our reinforcement learning experiments, we find that a default 50% flow ratio works robustly across a wide range of environments.
>
> > Related changes in the revision:
> > + Add Appendix J. Evaluating OFQL in High-Dimensional Action Robotic Manipulation
> > + Existing content: Ablation Study/Ablation on flow ratio
>
> **Typo: citation in line 151.** We fixed the citation typo.
>
> ----
> Thank you again for your constructive comments and meticulous review, which have greatly helped us improve the quality of the paper. If anything in our response remains unclear, we would be happy to provide further clarification.

---

### Official Review · Reviewer_opYF · 2025-11-03

**Soundness:** 3
**Presentation:** 3
**Contribution:** 3
**Rating:** 6
**Confidence:** 4

**Summary:**

To overcome the limitations of DQL, the paper proposes replacing the multi-step denoising policy used in training and inference with a one-step denoising policy. Unlike other one-step approaches that require an auxiliary teacher network for distillation, the paper adopts a mean-flow policy that directly approximates the denoising process. The proposed method demonstrates strong empirical performance and improved efficiency.

**Strengths:**

The method is simple, clear, and effective. By replacing only the diffusion policy component with the mean-flow policy, the approach achieves both higher sampling efficiency and competitive performance. The toy example nicely illustrates the advantage of reparameterizing from $v$ to $u$, providing a clearer intuition for the underlying mechanism.

**Weaknesses:**

Given that mean-flow generative modeling has already shown strong one-step FID results on image generation tasks, it would be valuable to see this approach applied to more complex environments beyond D4RL, such as robotic control or high-dimensional decision-making settings.

**Questions:**

The model performs worse on the Kitchen and AntMaze-Large-Diverse tasks, which are relatively more challenging within the D4RL benchmark. Do the authors have any insights into these results?
Could it be that the mean-flow policy limits exploration during training, leading to reduced performance on tasks requiring greater stochasticity?

---

> ### Author Response · Authors · 2025-11-26
> **Rebuttal by Authors**
>
> Thank you for the detailed review and constructive feedback on this work.
> Below, we provide answers to your questions and present additional experiments aimed at addressing the identified weakness.
>
> **Question: Insight about AntMaze-Large-Diverse and Kitchen-Mixed results and the exploration of OFQL policy.**
>
> We appreciate the reviewer for raising this subtle but important point. Kitchen-Mixed and AntMaze-Large-Diverse are stitching-heavy tasks: the agent must compose short, fragmented sub-trajectories into a coherent long-horizon solution, even though the offline dataset almost never contains complete successful trajectories for the full task. Success therefore critically depends on (1) strong generalization of the Q-function to unseen intermediate states lying along feasible stitching paths and (2) accurate long-horizon credit assignment derived from such generalization.
>
> Our method (QFQL) primarily enhances policy learning by providing a fast and expressive actor update, while the critic is still trained using standard TD learning on the offline dataset. In stitching-heavy settings such as Kitchen-Mixed and AntMaze-Large, TD learning alone may struggle to extrapolate reliable values for unseen stitching states. As a result, the policy—regardless of its expressivity or stochasticity—may not receive sufficiently informative Q-value gradients to guide effective long-horizon behaviors.
>
> While the reviewer’s hypothesis about policies limiting exploration is highly relevant in online RL, in the offline regime we believe the bottleneck here is critic-centric: When the Q-function fails to provide informative estimates beyond the dataset’s support, increased policy expressivity or stochasticity cannot meaningfully enhance trajectory stitching, and thus has limited impact on overall task performance.
>
> This insightful question also points toward a promising research direction: enhancing the stitching and long-horizon credit assignment capabilities of the critic within the QFQL framework. Strengthening Q-function generalization to feasible stitching paths could improve performance on these challenging domains. We consider this a valuable and non-trivial extension for future work
>
> **Addressing Weakness: Experiments on Robotic Control and High-Dimensional Decision-Making**
>
> **High-dimensional action.** To further evaluate OFQL in high-dimensional action spaces, we conducted additional experiments on the D4RL Adroit benchmark, , which features 24-dimensional control using a dexterous robotic hand.We evaluated two standard tasks—adroit-pen-human and adroit-pen-cloned—where the objective is to manipulate a pen to match a target orientation using a 24-DoF hand. This domain is particularly challenging due to noisy human demonstrations, sparse rewards, and the high-dimensional action manifold. Normalized returns are reported below:
> ||BC|DQL|OFQL|
> |--|--|--|--|
> |adroit-pen-human|71.0|75.7+-9.0|**79.5+-9.5**
> |adroit-pen-cloned|52.0|60.8+-11.8|**62.3+-10.3**
>
> The results show that OFQL consistently outperforms both BC and DQL in
> high-dimensional manipulation tasks, demonstrating strong robustness and
> effectiveness in complex dexterous control settings.
>
> **High-dimensional observation.**  We additionally evaluate OFQL in high-dimensional input settings using the OGBench benchmark (2 visual tasks). The results show that OFQL remains functional with raw visual observations (64\*64\*3 image); however, its performance is still limited. We argue that achieving strong results in visual offline RL likely requires additional architectural and training design, such as stronger visual encoders, improved conditioning strategies, or representation regularization.
> |  |FBRAC|FQL|OFQL|
> |--|--|--|--|
> |visual-scene-play-singletask-task1-v0|46.0+-4.0|**98.0+-3.0**| 54+-9.0|
> |visual-cube-double-play-singletask-task1-v0|6.0+-2.0|**21.0+-11.0**|8.0+-3.0|
>
> We hope these results further strengthen the paper and provide readers with a more comprehensive understanding of OFQL’s applicability across both high-dimensional action and visual input domains.
>
> For detail information, please refer the revision.
>
> > Changes Added in the Revision:
> > + Add Appendix J. Evaluating OFQL in High-Dimensional Action Robotic Manipulation
> > + Add Appendix K. Feasibility on visual observation
>
> ---
>
> We once again sincerely thank the reviewer for the time spent on the review and for raising an insightful question. We hope that our clarifications adequately address the identified weakness, and we would be grateful if you would consider this in your updated evaluation.

---

### Public Comment · ~Flow_King1 · 2025-11-12
**Concerns on Experimental Fairness and Potential Bias in Baseline (FQL) Comparison**

1. **Experimental reliability.** The experimental results reported in this paper appear unreliable. OFQL is evaluated on several environments that are not part of the official FQL testbed, which undermines the fairness of the comparison. The authors should run on `ogbench` to ensure parity. In addition, they should release all hyperparameters and, ideally, the Weights & Biases (wandb) logs to substantiate the findings. For example, on `HalfCheetah-Medium-Expert`, with $\alpha \in {3, 10, 100, 200}$, we readily obtain a normalized score of $98.6 \pm 0.8$, with similar trends on other tasks. We therefore recommend either (i) using the same evaluation suite as FQL, or (ii) publicly disclosing key hyperparameters (e.g., `alpha`) and logs to clarify the source of performance gains.

2. **Positioning relative to FQL.** This work is closely related to FQL rather than DQL, which is also a one-step flow Q-learning method; therefore, the title “Addressing the Diffusion Policy Bottleneck” may be misleading, since FQL likewise employs a one-step Q-learning formulation. The paper should provide a deeper comparison with FQL—e.g., explaining why a MeanFlow-based, inherently one-step approach would be preferable to FQL’s distillation-based one-step design.

This review is entirely objective, carries no conflict of interest, and is made solely in the interest of promoting a healthier and more constructive development of the academic community.

---

> ### Comment · Area_Chair_hJia · 2025-11-15
> **Violation of ICLR 2026 Author Guide by This Comment - From AC**
>
> This comment is **INVALID** as it violates the ICLR 2026 Author Guide. Therefore, readers should not be influenced by this comment.
>
> According to the guidelines, "*OpenReview will allow for public discussion at any time during the discussion phase. Anyone who is logged in can post comments that are either publicly visible or restricted to reviewers, area chairs (ACs), or program committee (PC) members. All comments, except those from authors, reviewers, ACs, or the organizing committee, must be **non-anonymous**.*"
>
> If Flow King wishes to raise this issue, please do so in a non-anonymous manner.
>
> AC

---

> ### Public Comment · ~Flow_King1 · 2025-11-15
> **Just a Friendly Reminder — Anonymous.**
>
> I strongly agreed with AC.  My comment is just a **friendly reminder**, and this is my final comment.
>
> I strongly encourage readers and reviewers to consult prior work and carefully compare the experimental settings, datasets, and results with those in this paper. In reinforcement learning, it is quite easy to obtain relatively weak baselines, so using consistent experimental setups is very important—this is why most follow-up works to FQL report results on **OGBench**. If readers are interested in this discussion, they may also reproduce FQL under different values of `alpha` to further examine the effect of this parameter.
>
> I fully respect the ICLR 2026 Author Guide. In an environment where the anonymity of peer review is increasingly fragile and comments can easily be disseminated on social media, I reluctantly chose this approach. However, out of respect for the academic community and a desire to uphold its norms, I raised this issue anonymously. I hope this will not affect the review of the paper, but it may be helpful for readers who intend to follow or build upon this line of work.
>
>
>
>
> [1] Park S, Li Q, Levine S. Flow q-learning[J]. arXiv preprint arXiv:2502.02538, 2025. (ICML 2025)
> [2] Espinosa-Dice N, Zhang Y, Chen Y, et al. Scaling Offline RL via Efficient and Expressive Shortcut Models[J]. arXiv preprint arXiv:2505.22866, 2025.
> [3] Park S, Frans K, Eysenbach B, et al. Ogbench: Benchmarking offline goal-conditioned rl[J]. arXiv preprint arXiv:2410.20092, 2024. (ICLR 2025)
> [4] Jang Y, Nam H C, Park J M, et al. Q-Guided Flow Q-Learning[C]//CoRL 2025 Workshop RemembeRL.

---

> ### Author Response · Authors · 2025-11-26
> **Rebuttal by Authors**
>
> We provide the following clarification in response to your concerns.
>
> **Position of OFQL.** The reader’s comments appear to stem from an overemphasis on FQL and a misunderstanding of OFQL’s actual motivation and scope. OFQL is not a variant of FQL, nor is it designed to modify or improve FQL’s particular formulation. Instead, our research is motivated by a specific limitation of diffusion-based policies that require iterative denoising, which is inefficient in inference, training and requires backpropagation-through-time. In the landscape of solutions that tackle this inefficiency—such as distillation approaches (e.g, FQL) , efficient solvers (e.g., EDP), and alternative training–inference pipelines (e.g., IDQL, DD, ..)—OFQL is positioned as a simple, more direct one-step training–inference pipeline.
>
> The method's name, OFQL, is chosen purely as a **memorable identifier** that reflects our approach. It is not intended to suggest that we extend, improve, or alter FQL, nor that we convert FQL from "one-step" into a different "one-step" method. Rather, OFQL stands alongside FQL and other methods, providing practitioners with an additional, effective choice depending on their use case. We structured the paper to make this positioning explicit, and we hope this clarification resolves any concern regarding the name, title, or its implications.
>
> **Experimental Reliability.** The reader's concern appears to stem from a misleading observation based on a finely tuned, non-recommended FQL configuration that produces an isolated gain on one dataset but fails to generalize and becomes impractical when applied across many environments.
>
> - **FQL Hyperparameters.** We investigated the suggested improvement on HalfCheetah-Medium-Expert. Our findings: We **followed the official FQL highly recommended approach**: using the normalized Q option and searching alpha in (0.03,0.1,0.3,1,3,10) as stated in Appendix C of the FQL paper/official GitHub. The reader instead used FQL without normalized Q and searched in a much larger range [3...200]. We attempted your approach and search on 9 locomotion tasks. While we observed improvements in HalfCheetah-medium-expert, most environments showed noticeable performance degradation (e.g., HalfCheetah-MR max 45; Hopper-ME max 71; Hopper-M max 70; Hopper-MR max 47, Walk2D-M max 60....) compared to our approach's results.
>
>    Optimal Alpha ranges vary widely without normalized Q in FQL (e.g., alpha can be 3,10 on AntMaze or up to 1000 for Relocate, and even 30,000 for Hammer). Conducting a fully exhaustive search across all datasets is challenging and inconsistent with the FQL authors’ recommended practice. We therefore favor the officially recommended configuration, while we do not deny that an exhaustive hyper-parameter search might yield additional gains in isolated settings.
>
> - **Reproducing Baselines:** To ensure fairness, we try our best to report results from original papers (e.g., DQL, FQL AntMaze results) or from broadly accepted reimplementations such as CleanDiffuser. For baselines whose results are not available (e.g., FQL on Locomotion), we reimplemented them carefully and reported the most reliable results we could reproduce.
>
> **Choice of Benchmark.** We favor D4RL as it (1) has diverse task families and dataset qualities with clear benchmark properties (e.g, undirected, narrow and biased, non-markovian ...), (2) is the most widely accepted benchmark for offline RL (3) provides reliable, well-documented baseline results for all major algorithms we compare against.  These characteristics make D4RL the most suitable benchmark for evaluating and understanding OFQL’s behavior (See more details in Appendix/Benchmark task). We acknowledge that OGBench is a promising newer benchmark, originally designed for **goal-conditioned tasks** though. However, the benchmark is not yet widely used for the specific set of algorithms we evaluate. Also, baseline results for key methods such as DQL are not currently available. Nevertheless, we appreciate your suggestion and will consider it for our vision-based extension.

---

### Author Response · Authors · 2025-11-28
**Summary of Revisions**

Dear PCs, SACs, ACs, and Reviewers,

We appreciate the reviewer’s constructive and insightful feedback. In response to the valuable comments provided, we have made the following adjustments to the paper. The technical revisions are highlighted in blue.


**Main revisions:**

> 1. Add a summary, explaining the formal justifications about average velocity learning, in the main paper (Design One-Step Policy) and link these to the formal justifications in Appendices G, H, and O
> 2. Add Appendix G. FORMAL JUSTIFICATION OF ACTION ACCURACY PRESERVATION IN ONE-STEP GENERATION THROUGH AVERAGE VELOCITY FIELD
> 3. Add Appendix H. FORMAL JUSTIFICATION OF AVERAGE-VELOCITY LEARNING ENCOURAGES THE LEARNED ONE-STEP POLICY TO STAY CLOSE TO THE BEHAVIOR POLICY
> 4. Add Appendix I. GRADIENT ANALYSIS OF OFQL ACTOR LOSS
> 5. Add Appendix J. EVALUATING OFQL IN HIGH-DIMENSIONAL ACTION ROBOTIC MANIPULATION
> 6. Add Appendix K. FEASIBILITY ON VISUAL OBSERVATION SETTING
> 7. Add appendix L. HANDLING OUT-OF-DISTRIBUTION STATES: OFQL vs. DQL
> 8. Add appendix M. ABLATION ON TIME-SAMPLING DISTRIBUTION
> 9. Add Appendix N. BASELINES REPRODUCIBILITY to detail diffusion steps and how to get baseline results.
> 10. Update Appendix C to add an experiment comparing the expressivity of the average-velocity model and a DDPM baseline on controlled toy examples.



**Other minor revisions:**
> 1. Update the main paper to improve clarity: (lines 261-265) to improve the flow matching explanation on one-step generation; (line 323) flow ratio.
> 2. Fix the citation typos. (line 151).
---
Once again, thank you for the time and thoughtful reviews you devoted to this paper. Your efforts have made this paper stronger.

Best Regards,

Authors

---

### Meta-Review · Area_Chair_4VBY · 2026-01-06

**Summary:**

Overall, the paper proposes OFQL, which replaces the multi-step diffusion policy in DQL-style offline RL with a one-step mean-flow / average-velocity formulation so that action generation and actor optimization avoid iterative denoising and backpropagation-through-time. Reviewers generally found the method simple, clear, and empirically strong on D4RL, with notable efficiency gains.

The main concerns that informed my decision were:

• Novelty / positioning: whether OFQL is a sufficiently novel contribution beyond “swap diffusion loss for MeanFlow,” and whether the framing relative to DQL/FQL is accurate.

• Evaluation breadth and realism: results were initially concentrated on state-based D4RL; reviewers requested evidence on high-dimensional action and visual observation settings.

• Theoretical clarity: requests for a more formal explanation of why average-velocity learning enables accurate one-step generation and whether it constrains expressivity.

• Practicality / compute: whether the JVP term introduces meaningful overhead at scale.

• Weak-task behavior: understanding lower performance on Kitchen and AntMaze-Large-Diverse (trajectory stitching / long-horizon credit assignment).

**Reviewer Concerns:**

Mostly addressed:

• Baseline reproducibility details (diffusion steps, settings): The authors clarified step counts for major baselines and added a reproducibility appendix detailing baseline setting.

• High-dimensional action evaluation: Added D4RL Adroit (24-DoF) results showing OFQL remains competitive and improves over BC/DQL.

• Visual observation feasibility: Added initial OGBench visual task experiments and discussed challenges (encoder/conditioning/representation coupling).

• Theoretical justification / expressivity: Added formal arguments (appendices) connecting average velocity to the endpoint map and providing a justification that expressivity is not inherently reduced; also added toy experiments comparing average-velocity vs DDPM-style generation.

• Gradient interaction: Added gradient analysis clarifying how behavior regularization and return maximization terms interact.

• JVP overhead clarification: Provided an explanation that the JVP is a lightweight additional backward pass and included a scaling datapoint on a larger model.


Partially addressed:

• Depth of novelty: While the authors clarified that “minimal change” is intentional and non-trivial in RL, some readers may still view the contribution as primarily an adaptation of mean-flow parameterization to offline RL, rather than a fundamentally new RL algorithmic principle. The additional analyses help, but novelty perception remains mixed.

• Visual performance competitiveness: The added visual results demonstrate feasibility but do not yet match strong visual baselines in harder tasks. The paper now acknowledges this clearly; however, it remains an open limitation.

• Hard long-horizon / stitching tasks (Kitchen, AntMaze-Large-Diverse): The rebuttal provides a plausible critic-centric explanation (Q generalization / stitching), but this is not fully resolved with new algorithmic components. It remains a known weak spot and a future-work direction.

**Reviewer Scores:**

• 8FUi: no change

• opYF: no change

• 43m1: no change

• QtuL: no change

---

### Decision · Program_Chairs · 2026-01-26

Accept (Poster)